# Physiological modulation of BiP activity by *trans*-protomer engagement of the interdomain linker

**Steffen Preissler[1]\*, Joseph E Chambers[1], Ana Crespillo-Casado[1], Edward Avezov[1], Elena Miranda[2], Juan Perez[3], Linda M Hendershot[4], Heather P Harding[1], David Ron[1,5]\***

[1]Cambridge Institute for Medical Research, University of Cambridge, Cambridge, United Kingdom; [2]Department of Biology and Biotechnology, Charles Darwin, Sapienza University of Rome, Rome, Italy; [3]Laboratorio de Fisiología, Facultad de Ciencias, Universidad de Málaga, Málaga, Spain; [4]Department of Tumor Cell Biology, St. Jude Children's Research Hospital, Memphis, United States; [5]Wellcome Trust-MRC Institute of Metabolic Science, NIHR Cambridge Biomedical Research Centre, University of Cambridge, Cambridge, United Kingdom

**Abstract** DnaK/Hsp70 chaperones form oligomers of poorly understood structure and functional significance. Site-specific proteolysis and crosslinking were used to probe the architecture of oligomers formed by the endoplasmic reticulum (ER) Hsp70, BiP. These were found to consist of adjacent protomers engaging the interdomain linker of one molecule in the substrate binding site of another, attenuating the chaperone function of oligomeric BiP. Native gel electrophoresis revealed a rapidly-modulated reciprocal relationship between the burden of unfolded proteins and BiP oligomers and slower equilibration between oligomers and inactive, covalently-modified BiP. Lumenal ER calcium depletion caused rapid oligomerization of mammalian BiP and a coincidental diminution in substrate binding, pointing to the relative inertness of the oligomers. Thus, equilibration between inactive oligomers and active monomeric BiP is poised to buffer fluctuations in ER unfolded protein load on a rapid timescale attainable neither by inter-conversion of active and covalently-modified BiP nor by the conventional unfolded protein response.

**\*For correspondence:** sp693@cam.ac.uk (SP); dr360@medschl.cam.ac.uk (DR)

## Introduction

In eukaryotes, the endoplasmic reticulum (ER) processes the vast majority of newly synthesized proteins destined for secretion and membrane insertion. The maturation of ER client proteins entails the chaperone assisted folding of nascent chains and assembly of functional complexes. Protein folding homeostasis in the ER requires that the complement of chaperones match closely the burden of nascent chains: too few chaperones risks misfolding and proteotoxicity (*Balch et al., 2008*), whereas too many chaperones slow down folding, impede cell growth and incur a cost of excessive client protein degradation (*Dorner et al., 1992*; *Feder et al., 1992*). The challenge in the ER is especially great, as the compartment experiences wide, physiologically-driven fluctuations in client protein load (*Itoh and Okamoto, 1980*; *Logothetopoulos and Jain, 1980*). But this dynamism also renders the ER a useful window into general principles that maintain protein folding homeostasis.

The importance of transcriptional regulation of ER chaperone-encoding genes has long been recognized (*Chang et al., 1989*) and the signal transduction pathways that couple unfolded protein

**eLife digest** Proteins are composed of long chains of amino acids that fold on themselves to form three-dimensional structures. Many proteins are made in a compartment within the cell called the endoplasmic reticulum and 'chaperone' proteins help them fold correctly. Cells carefully regulate the levels of chaperone proteins. If there are too few chaperones in the cell, then newly-made proteins may fold incorrectly and interrupt other processes. On the other hand, if too many chaperones are present they may slow down the protein folding process.

If a cell experiences stressful conditions, or if there is a sudden demand for more proteins to be made, protein folding can be disrupted. This leads to an increase in the number of unfolded proteins in the endoplasmic reticulum and so the cell increases the levels of chaperone proteins to cope with this.

Hsp70 chaperones are one family of chaperone proteins. These proteins can be present in a cell as single molecules (monomers) or as a group of several chaperone molecules (oligomers). Previous research has suggested that the chaperone proteins in oligomers are inactive, but the oligomers may be rapidly broken down into monomers when the cell needs to fold more proteins. A region within the Hsp70 called the 'interdomain linker' is important for regulating the chaperone activity, but it is not clear if this is connected to the formation of oligomers.

Preissler et al. used biochemical techniques to study how an Hsp70 protein in the endoplasmic reticulum called BiP forms oligomers. The experiments show that oligomers form when the interdomain linker of one BiP molecule is bound to the region of an adjacent BiP molecule that is normally reserved for binding to unfolded proteins. The presence of the interdomain linker makes it more difficult for unfolded proteins to bind.

Further experiments challenged cells with chemicals that caused the number of unfolded proteins in the cells to increase. Under these conditions, there was a decrease in the number of BiP molecules associated with oligomers. Preissler et al. propose that BiP oligomers act as reservoirs to store BiP molecules when they aren't needed by the cell. However, when the levels of unfolded proteins rise, cells can rapidly break up these oligomers to make active monomers that help to deal with the excess numbers of unfolded proteins. Further work is needed to understand how changes in the number of unfolded proteins in cells leads to the formation and disassembly of BiP oligomers.

stress (ER stress) to gene expression—collectively referred to as the unfolded protein response (UPR)—have been worked out in some detail (*Walter and Ron, 2011*). The inherent latency of this transcriptional program has favored the evolution of an additional strand to the UPR whereby, in animal cells, ER stress is coupled to rapid attenuation of protein synthesis (*Harding et al., 1999*). Whilst translation attenuation is an effective bulwark against misfolding and proteotoxicity, it caps the biosynthetic yield on ER activity (*Trusina et al., 2008*).

The inherent limitations and costs of the conventional transcriptional and translational UPR have generated interest in post-translational adaptations with shorter latency that rapidly match chaperone availability to client protein load. One such adaptation, apparently unique to the ER, is the reversible ADP ribosylation and de-ribosylation of the ER Hsp70 chaperone, BiP (*Carlsson and Lazarides, 1983*). In its ADP ribosylated form, BiP is locked into an inactive state with low affinity for its substrates (*Freiden et al., 1992*). ADP ribosylation affects residues in BiP that interfere with the allosteric coupling of ATP binding and hydrolysis in BiP's nucleotide binding domain (NBD) to the affinity of its substrate binding domain (SBD) for unfolded proteins (*Chambers et al., 2012*), thus targeting the basic chaperoning mechanism used by all Hsp70/DnaK family members (*Mayer and Bukau, 2005*).

A recent study reveals that the same circumstances that promote ADP ribosylation of BiP are also associated with rapidly reversible formation of an ester bond between threonine 366 in the NBD and adenosine monophosphate to yield BiP AMPylation, an inactivation modification (*Ham et al., 2014*). Whilst the interrelation of ADP ribosylation and AMPylation remain to be worked out, covalently modified BiP is understood to represent an inactive buffer that can be drawn upon in time of need, by removal of the modifications, thereby increasing the capacity of the ER to cope with enhanced unfolded protein burden whilst minimizing the cost of over-chaperoning. Indeed, the inter-conversion of ADP ribosylated and AMPylated BiP to the unmodified active form follows a coherent pattern

whereby increased client protein load converts the inactive buffer form to the active one and diminished client protein load pushes modification the other way (*Laitusis et al., 1999*; *Ham et al., 2014*; *Sanyal et al., 2015*).

The transitions noted above occur over a time scale of ~1 hr, allowing for more rapid adaptations. One such potential adaptation entails oligomerization. Oligomerization of BiP, mammalian Hsc70 and bacterial DnaK, has been observed in vitro (*Blond-Elguindi et al., 1993*; *Benaroudj et al., 1995*; *King et al., 1995*; *Schonfeld et al., 1995*; *Aprile et al., 2013*) and in cells (*Freiden et al., 1992*; *Thompson et al., 2012*). The oligomers have attenuated client binding capability and may convert to the more active monomer (*Freiden et al., 1992*; *Blond-Elguindi et al., 1993*; *Chevalier et al., 1998*; *Thompson et al., 2012*; *Aprile et al., 2013*), suggesting that they may represent a rapidly accessible, inactive buffer form of chaperone. However, the molecular basis for oligomerization has remained obscure.

In recent years it has become apparent that the highly conserved interdomain linker connecting the NBD and SBD plays a crucial role in allosteric regulation of Hsp70 proteins (*Swain et al., 2007*; *Kityk et al., 2012*; *Qi et al., 2013*). Specifically, the aliphatic side chains of the conserved linker are reversibly buried against the NBD (in the ATP-bound state of the chaperone) or exposed to the solvent (in the ADP-bound state) mediating key allosteric transitions (*Bertelsen et al., 2009*; *Kumar et al., 2011*). Here we report on application of a chemical probe specific to the interdomain linker of BiP and site-specific crosslinking to analyze its disposition in BiP monomers and oligomers. In doing so we have uncovered a mechanism for rapid and reversible BiP oligomerization that inactivates the chaperone to serve as a recoupable storage form.

## Results

### In vitro BiP oligomerization by cross-protomer engagement of the interdomain linker at *trans* substrate binding sites

Allosteric coupling between the domains of DnaK/Hsp70-type chaperones is modulated by nucleotide binding to the regulatory NBD. In the ATP-bound state the two domains are in close juxtaposition and the side-chains of the hydrophobic interdomain linker are engaged at the interface of the NBD and SBD (*Liu and Hendrickson, 2007*; *Kumar et al., 2011*; *Kityk et al., 2012*; *Qi et al., 2013*), whereas in the ADP-bound state the domains are uncoupled and the interdomain linker is exposed to the solvent (*Swain et al., 2007*; *Bertelsen et al., 2009*). It therefore came as surprise to find that in vitro hydrolysis of the ATP in cell lysates (and its conversion to ADP) left a substantial fraction of BiP protected from digestion by a specific protease (SubA) that cleaves its interdomain linker (*Paton et al., 2006*) (*Figure 1A*).

These observations, made by manipulating the ATP content of cell lysates in vitro, are subject to the potential confounding effect of bound substrates, stabilized by ATP depletion, which might sterically hinder access of the protease to the interdomain linker. However, similar enhancement of linker exposure was also observed in pure preparations of BiP (*Figure 1B*), suggesting that an intrinsic aspect of BiP's biochemistry affords the linker protection under ATP depletion and enhances its exposure upon ATP repletion.

DnaK/Hsp70-type chaperones are prone to multimerization (*Freiden et al., 1992*; *Benaroudj et al., 1995*), which is enhanced in their ADP-bound state (*King et al., 1995*; *Schonfeld et al., 1995*). Crystal forms of a bacterial DnaK have been observed in which the interdomain linker of one protomer is engaged as a typical substrate by an adjacent protomer (*Chang et al., 2008*). Whilst the in vivo relevance of *trans*-protomer linker binding has not been explored, the structure suggests a plausible mechanism by which ATP hydrolysis to ADP, which locks BiP in the high substrate-affinity mode, might protect the interdomain linker from digestion by SubA.

The architecture of BiP oligomers suggested by the aforementioned observations (*Figure 1C*) makes certain testable predictions in regards to the behavior of mutant forms of BiP. To explore the potential role of linker engagement by the SBD in oligomerization, we compared the oligomeric state of wildtype BiP, expressed in and purified from *Escherichia coli*, with BiP$^{V461F}$, a mutation that interferes with substrate engagement at the SBD (*Laufen et al., 1999*; *Petrova et al., 2008*) or a four residue substitution that alters the hydrophobic nature of the interdomain linker (L$^{414}$VLL$^{417}$ to A$^{414}$DDA$^{417}$, referred to as BiP$^{ADDA}$ henceforth) abolishing the linker's role in allostery (*Laufen et al., 1999*), disrupting the SubA cleavage site (*Paton et al., 2006*) and altering key residues that render the linker a BiP/Hsp70 substrate (*Chang et al., 2008*), all whilst preserving the intrinsic ability of BiP's SBD

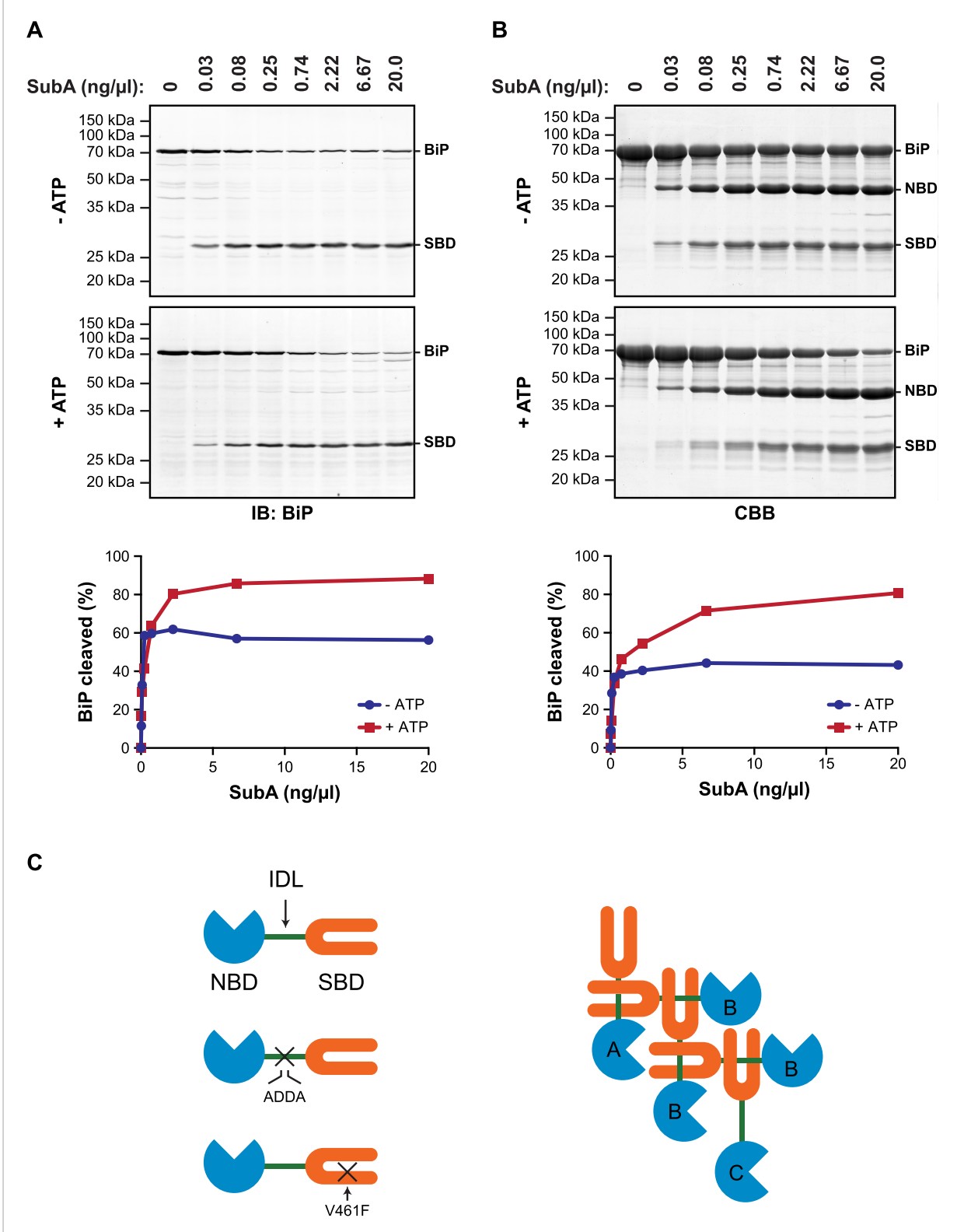

**Figure 1**. ATP increases sensitivity of BiP to cleavage at its interdomain linker. (**A**) Immunoblot of endogenous BiP in lysates of CHO-K1 cells, following exposure (10 min at 30˚C) to the indicated concentration of a bacterial protease (SubA) that selectively cleaves the interdomain linker and separation by SDS-PAGE. The lysate in the upper panel was prepared in presence of hexokinase and glucose to deplete ATP. The lysate in the lower panel was prepared without hexokinase and exposed to ATP (1.5 mM) before cleavage by SubA. The full-length protein (BiP) and the C-terminal substrate binding

*Figure 1. continued on next page*

Figure 1. Continued

domain (SBD) were detected with antiserum that recognizes a C-terminal epitope of hamster BiP, quantified by fluorescence-based immunoblotting on a Licor Odyssey imager and displayed in graphic form in the bottom panel. (**B**) Purified BiP (12 μM) was incubated (10 min at 30°C) with or without 3 mM ATP followed by exposure (20 min at 30°C) to the indicated concentration of SubA. The samples were then analyzed by SDS-PAGE and Coomassie (CBB)-staining. The intact protein (BiP), nucleotide binding domain (NBD) and SBD are indicated. Signal intensity was quantified by detecting Coomassie fluorescence on a Licor Odyssey imager and displayed in graphic form in the bottom panel. Shown are representative experimental observations, reproduced three times. (**C**) Cartoon describing the proposed architecture of BiP oligomers: The SBD is in orange, the NBD in blue and the interdomain linker (IDL) in green. The sites of the ADDA mutation that abolishes SubA cleavage and alters the character of the interdomain linker and that of the V461F mutation that interferes with substrate binding are indicated (crosses). The oligomer is predicted to encompass protomers with three different dispositions. Protomer A is allosterically uncoupled but possesses a free SBD capable of engaging substrates (or other BiP molecules through their interdomain linker). Protomer B is allosterically uncoupled and unable to bind substrates. Protomer C is unable to bind substrates but is subject to allosteric regulation via its NBD.

to bind other substrates. Size-exclusion chromatography (SEC) of purified wildtype BiP yields three discernable peaks (arbitrarily named I, II and III) with tailing into the high molecular weight region (*Figure 2A*, black trace), consistent with previous observations (*Chevalier et al., 1998*). The molar masses of BiP complexes in elution peaks I, II and III, as measured by multi-angle light scattering (SEC-MALS), match closely the predicted masses of BiP monomers, dimers and trimers, respectively (*Figure 2—figure supplement 1*). Addition of ATP leads to redistribution of BiP from the higher molecular weight fractions into a single large peak that elutes slightly later than peak I (*Figure 2B*). The late elution of the ATP-induced peak is consistent with linker-mediated compaction of the SBD and NBD (*Swain et al., 2007*).

In absence of nucleotide both the wildtype and the well-characterized $BiP^{T229A}$ mutant, which is able to bind ATP but fails to efficiently hydrolyze it (*McCarty and Walker, 1991*; *Gaut and Hendershot, 1993*; *Palleros et al., 1993*; *Wei et al., 1995*) (*Figure 2—figure supplement 2A*), formed oligomeric complexes that entirely disassembled upon addition of ATP (*Figure 2—figure supplement 2C,D*). In contrast, oligomers of wildtype BiP were largely maintained in presence of ADP, whereas ATP-γ-S, a slowly-hydrolyzable ATP analog, caused partial dissociation of the large oligomeric species. Together this suggests that the conformational change (and the concomitant decrease in substrate affinity) induced by ATP binding, rather than by its hydrolysis, triggers BiP oligomer dissociation.

$BiP^{V461F}$ elutes mostly in peak I and its elution is also slightly retarded by ATP (*Figure 2A*, purple trace and *Figure 2C*), pointing to intact allostery and structural integrity of the mutant. $BiP^{ADDA}$ elutes with less sharply-defined peaks. Like $BiP^{V461F}$ it too is preferentially distributed into the late-eluting peak I (*Figure 2A*, green trace), whereas the compound $BiP^{ADDA;V461F}$ mutant resembles the single $BiP^{V461F}$ mutant (*Figure 2A*, pink trace). Importantly, addition of ATP had no effect on the migration of $BiP^{ADDA}$ (*Figure 2D*), consistent with abolition of allosteric coupling between the domains by the alteration of the hydrophobic linker (*Laufen et al., 1999*; *Swain et al., 2007*; *Kumar et al., 2011*).

Competition for BiP's SBD either by a well-characterized BiP substrate peptide, (HTFPAVL [*Marcinowski et al., 2011*]), or a peptide corresponding in sequence to the interdomain linker (DTGDLVLLD), dissociated the higher molecular weight forms of BiP and increased intensity of peak I, whereas a peptide derived from the $BiP^{ADDA}$ mutant linker sequence (DTGDADDAD) had no effect (*Figure 2E*). The nucleotide and substrate peptide-mediated rearrangements of BiP complexes observed by SEC were mirrored by the migration of BiP on denaturing SDS-PAGE, following light crosslinking by short exposure to gluteraldehyde (*Figure 2—figure supplement 3*). Together, these experiments reveal that BiP oligomerization depends on integrity of both the SBD and the hydrophobic character of the interdomain linker and likely involves engagement of one BiP molecule by another as a typical substrate.

To further explore the architecture of BiP oligomers, we incubated wildtype BiP with trace amounts of a fluorescent, lucifer yellow-labeled BiP substrate peptide HTFPAVL (Pep-LY) (*Marcinowski et al., 2011*) and simultaneously recorded the protein quantity (by monitoring peptide bond absorbance at 230 nm) and fluorescent signal as the complex eluted from the SEC column. Incorporation of peptide as a function of BiP quantity fell progressively with increased complex size. The ratio of fluorescent to quantity signal ($R^{FLU/A230\ nm}$), arbitrarily set to 1.00 for peak I, was 0.48 for peak II and 0.26 for peak III (*Figure 2F*), suggesting that peaks I, II and III were constituted of progressively larger BiP oligomers

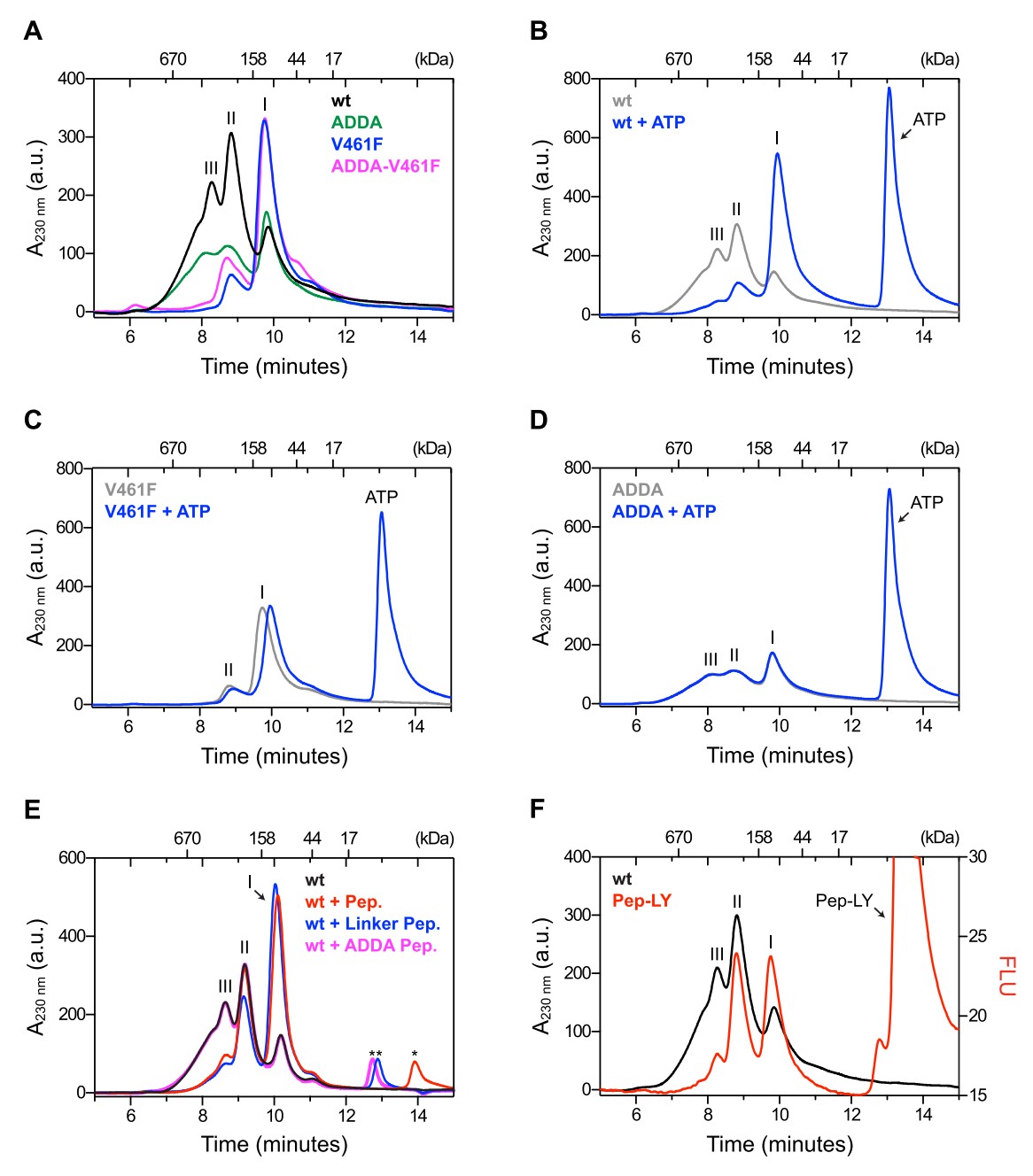

**Figure 2**. Mutations that disrupt substrate binding or alter the interdomain linker interfere with BiP oligomerization. (**A**) Peptide bond absorbance traces ($A_{230\ nm}$) of solutions of bacterially expressed and purified wildtype (wt) BiP, or the indicated mutants (all at 50 μM), following fractionation by size-exclusion chromatography (SEC-3 column). The distinct peaks eluting at 9.85 min (I), 8.82 min (II) and 8.28 min (III) are marked. The ADDA mutation ($L^{414}VLL^{417}$ to $A^{414}DDA^{417}$) inactivates the linker's role in interdomain allostery (*Swain et al., 2007*; *Kumar et al., 2011*), whereas the V461F mutation inhibits substrate binding (*Petrova et al., 2008*). Note relatively fewer oligomeric species (fractions III and II) in the preparations of mutant proteins. (**B**) Trace as in 'A' of wildtype BiP incubated in the absence or presence of 5 mM ATP before fractionation. The BiP profile described in 'A' and the profile of BiP incubated with ATP are shown. The additional peak at 13.07 min represents free ATP. Note the dissociation of oligomers (from peaks III and II to peak I) upon exposure to ATP. (**C**) Trace as in 'B' with the V461F mutant BiP. (**D**) Trace as in 'B' with the ADDA mutant BiP. (**E**) Wildtype BiP (50 μM) was incubated without or with a BiP-binding peptide (HTFPAVL, 'Pep.'; 1 mM) (*Marcinowski et al., 2011*) or a peptide derived from the interdomain linker of BiP (DTGDLVLLD, 'Linker Pep.'; 1 mM) or a mutant linker peptide (DTGDADDAD, 'ADDA Pep.'; 1 mM) and fractionated as in 'A'. Note that whereas addition of both the reference peptide (HTFPAVL) and the linker peptide (DTGDLVLLD) shifted the distribution of BiP to lower molecular weight species, the mutant linker peptide (DTGDADDAD) was with no effect. The asterisks mark the elution peaks of free peptides. (**F**) Wildtype BiP (50 μM) was incubated

*Figure 2. continued on next page*

*Figure 2. Continued*

in the absence or presence of tracer amounts lucifer yellow-labeled BiP-binding peptide (HTFPAVL-LY, 'Pep-LY'; 1 μM) and fractionated as in 'A'. Absorbance ($A_{230\ nm}$; black) and the fluorescence trace (in arbitrary fluorescence units, FLU) of lucifer yellow (LY) (Ex: 430 nm, Em: 525 nm; red) are shown.

The following figure supplements are available for figure 2:

**Figure supplement 1**. Multiangle light scattering (SEC-MALS) measurement of purified BiP eluted from a size-exclusion chromatography column.

**Figure supplement 2**. ATP binding rather than ATP hydrolysis induces dissociation of BiP oligomers.

**Figure supplement 3**. Glutaraldehyde crosslinking confirms the nucleotide dependence of BiP oligomerization and its susceptibility to competition by substrate peptides.

but with no concordant increase in peptide binding capacity. The latter finding is consistent with oligomers arranged such that the SBDs of all but the ultimate protomer are engaged in linker binding (*Figure 1C*).

Engagement of the interdomain linker from one protomer by the SBD of another could explain the resistance of BiP's interdomain linker to cleavage by SubA in the absence of ATP. In contrast, disruption of this homotypic chaperone:client complex upon exposure to ATP would result in enhanced accessibility of the linker to SubA. To examine this possibility in the context of BiP oligomers, we combined trace amounts of lucifer yellow (LY)-tagged BiP[V461F] (that is unable to engage substrates) with BiP[ADDA] that is resistant to cleavage by SubA (*Figure 3A*) and monitored fluorescence (which reports on the LY-BiP[V461F] tracer) and protein absorbance (which reports on mass, contributed largely by BiP[ADDA]) as the sample eluted from a SEC column. In the presence of BiP[ADDA], the LY-BiP[V461F] tracer distributed between a lower molecular weight peak I and higher molecular weight peak II consistent with a complex formed between BiP[ADDA] and the LY-BiP[V461F] tracer (*Figure 3B*). The latter was much diminished when BiP[ADDA] was replaced by the compound mutant BiP[ADDA;V461F], which is unable to engage substrate (*Figure 3C*). Exposure of the samples to SubA prior to SEC eliminated the fluorescent signal at peak I, replacing it with a later-eluting peak (tentatively labeled 'CP', for cleavage product) (*Figure 3B,C*, right panels). However, the BiP[ADDA]-dependent fluorescent peak II was notably resistant to SubA (*Figure 3B*), consistent with it arising from a molecule of LY-BiP[V461F] tracer whose (otherwise cleavable) interdomain linker is protected by engagement as a typical substrate by a BiP[ADDA] molecule, itself with an uncleavable linker.

It has recently been reported that oligomers of Hsp70 assemble by engagement of the interdomain linker with the C-terminal helical portion (the lid) of the SBD (*Aprile et al., 2013*). However, BiP, deleted of its helical lid (BiP[lidless]), still formed linker-dependent oligomers that dissociated upon exposure to ATP (*Figure 3—figure supplement 1A,B*), engaged peptides in a hierarchy similar to that of wildtype BiP oligomers (compare *Figure 2F* with *Figure 3—figure supplement 1C*) and protected the interdomain linker of LY-BiP[V461F] tracer similarly to wildtype BiP (*Figure 3—figure supplement 1D*). These observations point to the dispensability of the helical lid for BiP oligomerization via linker engagement and support the conclusion that the linker is engaged by the SBD as a typical substrate in a configuration consistent with the crystal structure of *G. kaustophilus* DnaK (*Chang et al., 2008*).

The aforementioned observations pointing to a mechanism of oligomerization that entails linker engagement suggest that ATP binding has two opposing effects on interdomain linker exposure: (*i*) Protection against SubA by burial at the interface of the coupled NBD and SBD; a state that equilibrates relatively rapidly with the uncoupled linker-exposed state, leading ultimately to complete digestion by SubA. (*ii*) Exposure to SubA by release of the linker from the SBD of adjacent protomers in the oligomer. In the absence of ATP the oligomers are very stable with a low off rate (*Figure 2—figure supplement 3B*). Thus the addition of SubA to the ATP-depleted sample leads to rapid cleavage of any wildtype BiP monomers (and of the exposed protomer C from oligomers, see cartoon in *Figure 1C*) with the persistence of a fraction of SubA resistant oligomeric BiP, which equilibrates very slowly with cleavage susceptible monomers. In case of the wildtype BiP, effect *ii* was

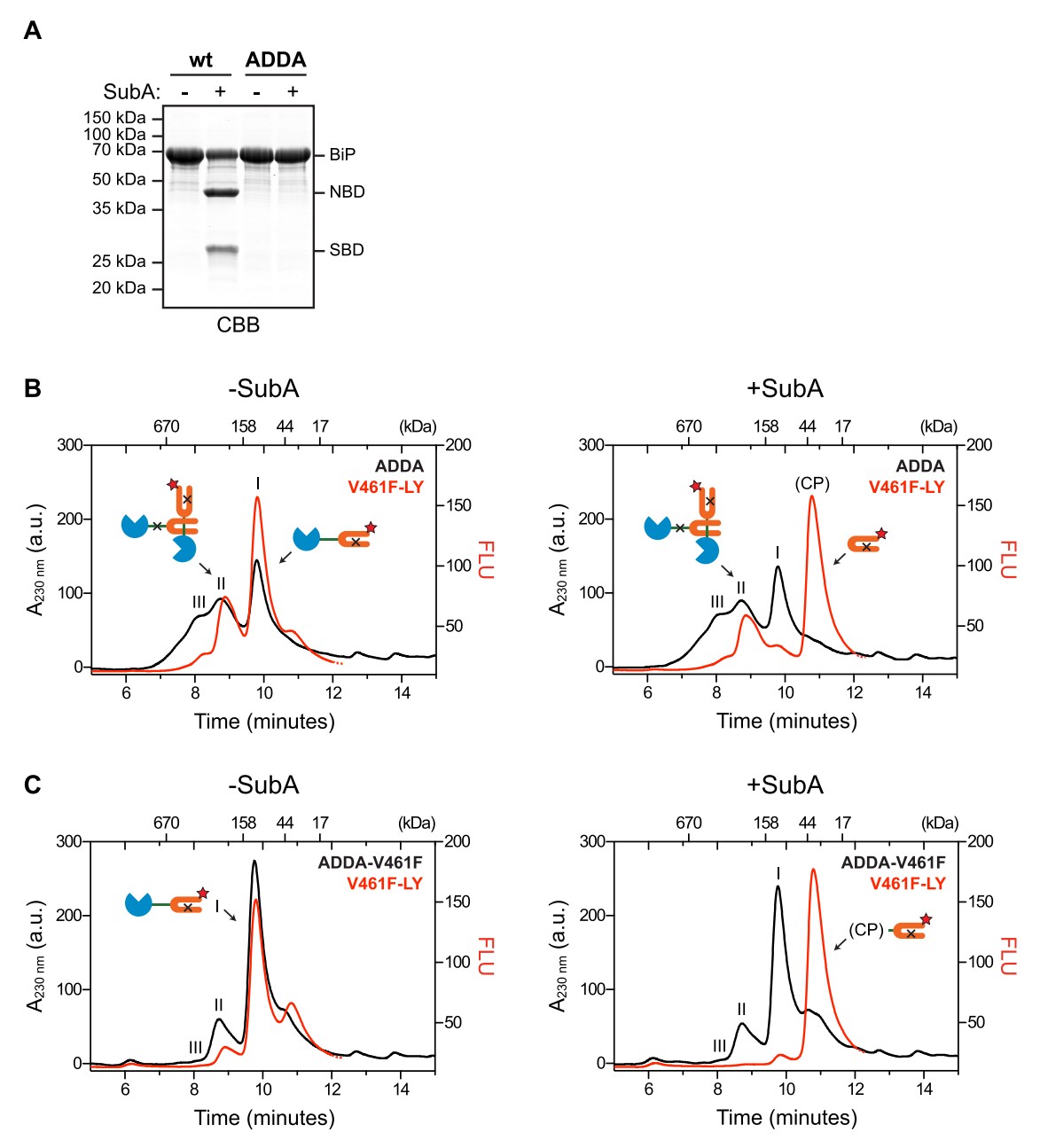

**Figure 3.** Cross-protomer engagement of the interdomain linker in BiP oligomers protects it against cleavage by SubA. (**A**) Coomassie-stained SDS-gel of wildtype (wt) and ADDA mutant BiP (12 µM) before and after cleavage with SubA (0.25 ng/µl) for 15 min. Note that the ADDA mutation precludes cleavage by SubA. (**B**) Peptide bond absorbance trace ($A_{230\ nm}$, black) and fluorescence trace (in arbitrary fluorescence units, FLU) of lucifer yellow (LY) (Ex: 430 nm, Em: 525 nm; red) of purified ADDA mutant BiP (50 µM) incubated with tracer concentrations (1 µM) of lucifer yellow-labeled V461F mutant BiP (V461F-LY) and fractionated by size-exclusion chromatography (as in *Figure 2*) either before (-SubA) or after (+SubA) cleavage with SubA (36 ng/µl). Peaks I, II and III are marked as is the new fluorescent peak emanating from the V461F-LY cleavage product (CP). Note that fluorescent peak I, corresponding to the V461F mutant BiP monomer is attenuated after cleavage with SubA, whereas peak II, arising from hetero-oligomers of ADDA mutant BiP and V461F mutant BiP is relatively resistant. The peptide-bond absorbance trace ($A_{230\ nm}$), contributed largely by the ADDA mutant BiP, is unchanged by cleavage with SubA. (**C**) Experiment as in 'B' with purified double mutant ADDA-V461F BiP (50 µM) and lucifer yellow-labeled V461F mutant BiP (V461F-LY). Note the absence of a SubA-resistant fluorescent peak II in this sample.

The following figure supplement is available for figure 3:

**Figure supplement 1**. Removal of BiP's lid does not abrogate cross-protomer engagement of the interdomain linker in BiP oligomers.

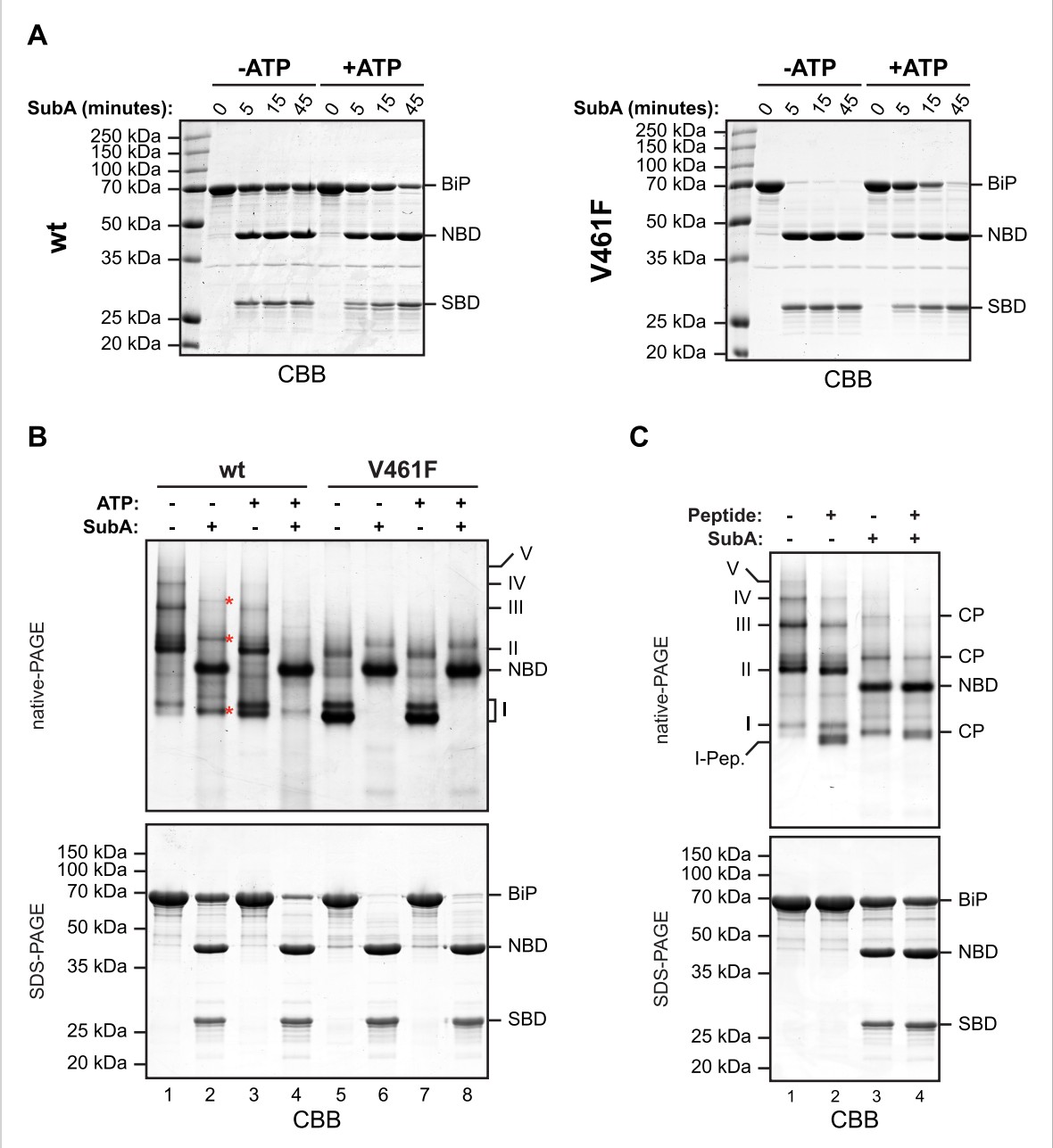

**Figure 4**. In oligomers, only a subset of protomers expose their interdomain linker to SubA. (**A**) Purified wildtype (wt) and V461F mutant BiP (both at 7.5 µM) were exposed to SubA (20 ng/µl) at 30°C for the indicated time in presence or absence of 2 mM ATP and analyzed by SDS-PAGE and Coomassie staining. The intact protein (BiP), nucleotide binding domain (NBD) and substrate binding domain (SBD) are indicated. Note that ATP enhances cleavage of the wildtype BiP whilst attenuating cleavage of the V461F mutant. (**B**) Coomassie-stained native gel (upper panel) of purified wildtype and V461F mutant BiP (both at 50 µM). Where indicated the sample had been exposed to 5 mM ATP and SubA (15 ng/µl) and digested to completion. The major species have been numbered I-V by order of descending mobility. Red asterisks mark the new species emerging after digestion with SubA and the cleaved NBD is noted for reference. The same samples were applied to SDS-PAGE and the gel was stained with Coomassie (lower panel). (**C**) Wildtype BiP (50 µM) was incubated in presence or absence of 500 µM BiP-binding peptide (HTFPAVL [*Marcinowski et al., 2011*]) for 16 hr at room-temperature followed by digestion with SubA, as is 'B'. The samples were then analyzed in parallel by native-PAGE (upper panel) and SDS-PAGE (lower panel) and the gels were stained with Coomassie. The novel species emerging after addition of peptide is labeled 'I-Pep' and the cleavage products of BiP (CP) are indicated. Note that both peptide addition and ATP shift the distribution of BiP oligomers to faster migrating species on native gels and enhance sensitivity to cleavage by SubA.

The following figure supplement is available for figure 4:

**Figure supplement 1**. BiP protomers have high affinity for each other.

seen to dominate, but in BiP$^{V461F}$ (that is unable to engage the linker in its SBD), effect $i$ dominates (*Figure 4A*). Of note, the divergent consequences of ATP addition to the two SubA cleavage reactions presented in *Figure 4A*, argues against an important role for nucleotide in SubA action.

Further support for oligomerization by *trans*-engagement of the interdomain linker in the SBD of an adjacent protomer was provided by native gel electrophoresis. Wildtype BiP formed discrete oligomers (labeled II, III, IV and V), whilst BiP$^{V461F}$ populated a dominant high mobility species (I) (*Figure 4B*). Exposure to SubA did not completely disassemble the oligomeric species, but rather converted them to a new series of higher mobility (compare lanes 1 and 2 in *Figure 4B*). However, in the presence of ATP, wildtype BiP was digested to near completion by SubA, with the NBD as the dominant remaining species (the SBD is not resolved on the native gel) (lane 4, *Figure 4B*); a pattern similar to that presented by BiP$^{V461F}$ in the presence or absence of ATP (*Figure 4B*, lanes 6 and 8). Peptide competition had a similar, though less pronounced effect as ATP, biasing the oligomeric species towards higher mobility forms that were converted to a new series of even higher mobility by SubA (*Figure 4C*).

Native gel electrophoresis also provided a crude estimate of the dissociation rate of the oligomers. The concentration of BiP at which half of species II ($K_{1/2max}{}^{II}$) and half of species III ($K_{1/2max}{}^{III}$) dissociated was 0.45 μM and 0.74 μM, respectively (*Figure 4—figure supplement 1*). Whilst incomplete re-equilibration and inter-conversion amongst oligomeric species likely render this an upper estimate of the affinity of BiP protomers for one another, nonetheless the observations point to considerable stability of the oligomers, as suggested previously (*Chevalier et al., 1998*).

The discrete oligomeric species observed in native gel electrophoresis of wildtype BiP were diminished in BiP$^{ADDA}$ (*Figure 5A*, compare lanes 1 and 3), as predicted from the behavior of this mutant on SEC (*Figure 3* and *Figure 3—figure supplement 1*). This observation is consistent with low affinity of the SBD for the mutant interdomain linker. However, the crystal structure of the ATP-bound homologous DnaK indicates that side chains of BiP linker residues affected by the ADDA mutation (L414 & L416) are directly involved in ATP-mediated coupling of the SBD and NBD (*Kityk et al., 2012*; *Qi et al., 2013*). Therefore, failure of BiP$^{ADDA}$ to oligomerize is also consistent with an allosteric effect of the mutation.

To address this experimentally we exploited the fact that BiP produces characteristic conformation-dependent proteolytic patterns in low concentrations of proteinase K (*Kassenbrock and Kelly, 1989*; *Wei et al., 1995*). Such limited proteolysis of wildtype BiP generated a protected fragment corresponding in mass to the compact NBD (~44 kDa) as well as an N-terminal 60 kDa fragment that accumulated only in presence of ATP (species 2 in *Figure 5—figure supplement 1* and *Figure 5—figure supplement 2*) and thus likely reflects protection of the linker due to its engagement in interdomain contacts. In contrast, exposure of BiP$^{ADDA}$ to the protease resulted in rapid, nucleotide-independent appearance of the 44 kDa fragment, confirming that the protein was indeed unable to adopt a domain-coupled conformation.

Next we compared oligomerization of the BiP$^{ADDA}$ with other BiP proteins carrying mutations known to favor uncoupling of the SBD and NBD. Both the BiP$^{T37G}$ and BiP$^{G226D}$ mutations that cannot attain the ATP-induced coupled conformation ([*Wei et al., 1995*] and *Figure 5—figure supplement 1* and *Figure 5—figure supplement 2*) promoted oligomerization, in sharp contrast to BiP$^{ADDA}$ (*Figure 5A*). As predicted, the oligomers formed by the BiP$^{T37G}$ and BiP$^{G226D}$ mutants were unaffected by ATP, in contrast to wildtype BiP (*Figure 5A*), confirming their mutant status. The resistance of oligomers formed by the BiP$^{T37G}$ and BiP$^{G226D}$ mutants to dissociation by ATP, correlated with protection of the interdomain linker from cleavage by SubA (full-length BiP in *Figure 5B*), consistent with *trans*-engagement of their interdomain linker as a substrate by their uncoupled SBD. Together, these observations separate the allosteric effects of the ADDA mutation from its effects on oligomerization and support that defective oligomerization of BiP$^{ADDA}$ is imposed in *cis*, by altered composition of the mutant interdomain linker. The defect of BiP$^{ADDA}$ in the formation of discrete oligomers, though conspicuous, is incomplete. Impaired linker binding likely favors lower affinity interactions that involve other elements of BiP and may partially dissociate during separation by native gel electrophoresis (or SEC). This may explain the diffuse signals observed for BiP$^{ADDA}$ on native gels (lanes 3 and 4 in *Figure 5A*).

To further examine the potential for *trans*-engagement of the interdomain linker in the SBD, we incorporated a non-natural amino acid (*p*-benzoyl-$_L$-phenylalanine; pBpa, whose side chain can form UV-induced crosslinks with adjacent carbons) into specific sites of bacterially-expressed BiP$^{V461F}$ (*Figure 6A*) and purified the derivatized proteins (*Figure 6—figure supplements 1A,B*). When

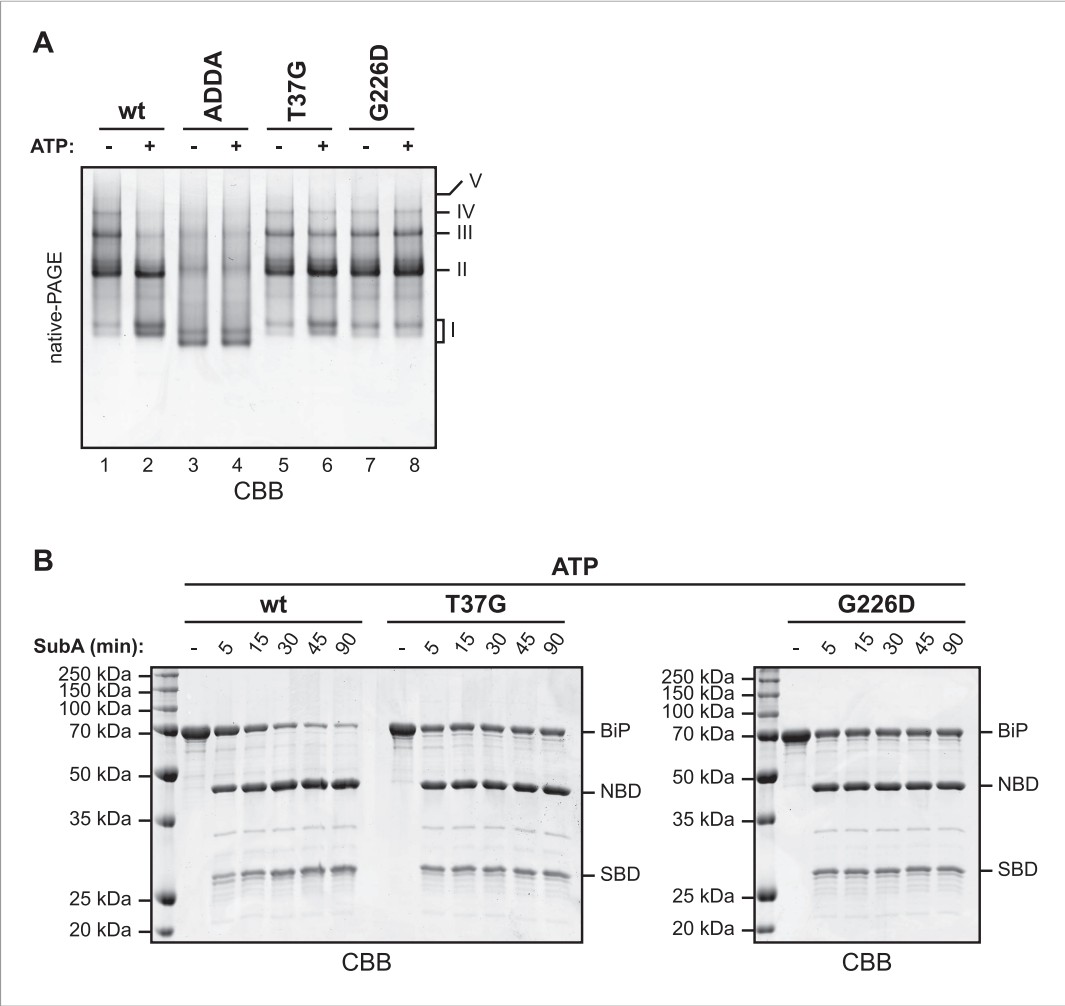

**Figure 5**. BiP[ADDA] fails to form discrete oligomers in contrast to other BiP mutants that are locked in the domain-uncoupled state. (**A**) Purified wildtype (wt) and mutant BiP proteins (BiP[ADDA], BiP[T37G] and BiP[G226D]) were incubated (all at 30 µM) for 15 min at 30°C with or without 3 mM ATP in HKM buffer and analyzed by native-PAGE and Coomassie-staining. The major species have been numbered I-V by order of descending mobility. Whereas BiP[ADDA] is severely impaired in the formation of discrete oligomers, the other mutants, whose nucleotide binding domains (NBD) and substrate binding-domains (SBD) are locked in the uncoupled state, form oligomers that persist in presence of ATP. (**B**) Purified wildtype and mutant BiP proteins (all at 7.5 µM) were exposed to SubA (20 ng/µl) at 30°C for the indicated times in presence of 3 mM ATP and subsequently analyzed by SDS-PAGE and Coomassie staining. The full-length proteins (BiP) as well as NBD and SBD are indicated. Note that the mutant BiP proteins are less efficiently cleaved by SubA.

The following figure supplements are available for figure 5:

**Figure supplement 1**. Partial proteolysis to analyze the nucleotide-dependent conformational states of wildtype and mutant BiP proteins.

**Figure supplement 2**. Characterization of the proteolytic fragments generated by digestion of BiP with Proteinase K.

combined with purified FLAG-tagged wildtype BiP, derivatives decorated with pBpa at positions located in the vicinity of the interdomain linker (L414, L416, L417, L419) resulted in a conspicuous, UV-dependent shift in the mobility of the unmodified FLAG-tagged BiP on denaturing SDS-PAGE (*Figure 6B*). Its aberrantly fast mobility likely reflects a compact conformation of the dimer. The novel

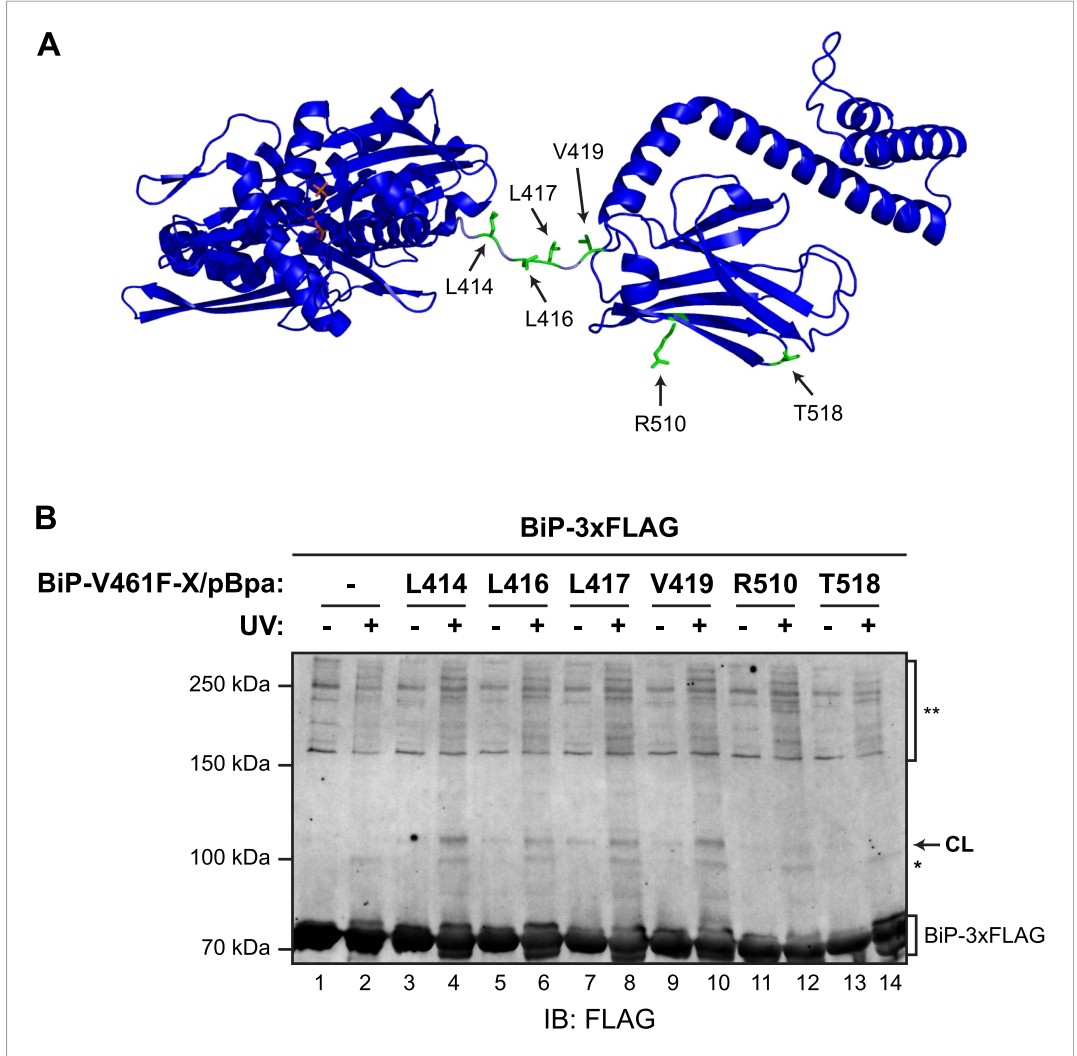

**Figure 6**. Site-specific photo-crosslinking to probe the trans-engagement of the interdomain linker in BiP oligomers. (**A**) Residues of BiP that were individually replaced by the photoactivatable crosslinking amino acid *p*-benzoyl-L-phenylalanine (pBpa) are highlighted (green) on a model of BiP in the ADP-bound state that is based on the crystal structure of DnaK from *Geobacillus kaustophilus* (PDB: 2V7Y). (**B**) Anti-FLAG immunoblot of FLAG-tagged wildtype BiP (BiP-3xFLAG) from in vitro photo-crosslinking reactions resolved on a 7% SDS-gel. BiP-3xFLAG (0.2 μM final) was combined with untagged BiP-V461F-X/pBpa proteins (2 μM final) carrying pBpa at either one of the positions (X) illustrated in 'A'. The mixtures were further incubated for 16 hr at 25°C to allow for efficient formation of *trans* BiP complexes. Where indicated samples were then exposed to UV light for 20 min on ice. The position of pBpa decorated BiP-dependent ('specific') crosslinked adduct is indicated (CL) as are the decorated BiP-independent ('non-specific adducts', *) and FLAG immunoreactive high-molecular weight material (**). Note that the specific UV-enhanced CL signals were diminished (or absent) when pBpa was incorporated at positions (aa 510 and 518) distant from the interdomain linker.

The following figure supplement is available for figure 6:

**Figure supplement 1**. Site-specific photo-crosslinking yields a specific adduct.

modified BiP binding partner-dependent species stood out against the background of UV-dependent changes in the mobility the FLAG-tagged BiP observed in absence of a pBpa-decorated partner; crosslinks to nucleotide likely contribute to altered mobility of UV-irradiated FLAG-tagged BiP during SDS-PAGE. By contrast, incorporation of pBpa at residues distant from the interdomain linker (R510 and T518) gave rise to a much weaker crosslink signal (***Figure 6B***, lanes 12 and 14

and (*Figure 6—figure supplement 1C*). These observations point to physical proximity of pBpa-substituted residues in the vicinity of the interdomain linker of untagged BiP$^{V461F}$ to its FLAG-tagged unmodified wildtype BiP counterpart, supporting the model of *trans*-protomer engagement of the interdomain linker cartooned in *Figure 1C*.

## Linker engagement at the SBD as the basis for BiP oligomerization in vivo

Native gel electrophoresis confirmed the presence of discrete higher molecular weight species of BiP in cell lysates, which stand out above the heterogeneous signal expected of a chaperone that engages many substrates [as previously noted, (*Freiden et al., 1992*)]. These were further stabilized by ATP depletion (*Figure 7A*). The migration of these oligomeric species on native gels was similar to that of untagged mature BiP (19–654), purified from bacteria (*Figure 7B*). Moreover, cleavage by SubA led to the appearance of a similar set of cleavage products, suggesting that the in vivo oligomers may also be comprised of one BiP protomer engaging the interdomain linker of another at its SBD.

To explore this possibility further, we exploited the observation that BiP$^{ADDA}$ was resistant to cleavage by SubA and was unable to serve as a substrate for linker engagement at the SBD. Plasmids encoding FLAG-tagged wildtype BiP or BiP$^{ADDA}$ were transfected into CHO-K1 cells and the migration of the exogenously introduced BiP and endogenous BiP were examined by native-PAGE immunoblot, before and after in vitro cleavage of the interdomain linker by SubA. To minimize the contribution of the exogenous BiP to the endogenous BiP signal on the immunoblot, we introduced an S649A mutation into the exogenous wildtype and ADDA mutant proteins. This converts residue 649 of the exogenous FLAG-tagged hamster BiP to that found in primates and eliminates its recognition by the R22 antiserum raised against the rodent sequence (*Hendershot et al., 1995*).

Both FLAG-tagged BiP$^{S649A}$ and FLAG-tagged BiP$^{ADDA;S649A}$ were incorporated into discrete high molecular weight species (*Figure 7C*). Exposure of the lysate from the cells transfected with FLAG-tagged BiP$^{S649A}$ to SubA yielded the familiar pattern of cleavage products in the anti-FLAG immunoblot (*Figure 7C*, lanes 3 and 4). The FLAG-tagged BiP$^{ADDA;S649A}$ was also incorporated into indistinguishable higher molecular weight species, however, unlike the endogenous BiP in the same lysate, the FLAG-tagged BiP$^{ADDA;S649A}$-containing oligomers were completely resistant to cleavage by SubA (*Figure 7C*, lanes 5 and 6). Given that BiP$^{ADDA}$ is defective in homo-oligomerization (*Figure 2A, D* and *Figure 7—figure supplement 1*), the oligomers observed in vivo were consistent with hetero-oligomers between endogenous BiP and the FLAG-tagged BiP$^{ADDA;S649A}$ in which the FLAG-tagged BiP$^{ADDA;S649A}$ is incorporated as a terminal protomer (protomer C in the schema of *Figure 1C*), protecting the interdomain linker of the endogenous BiP from digestion with SubA by engagement in the SBD of the BiP$^{ADDA;S649A}$ mutant.

Further support for oligomerization by linker engagement *in trans*, was provided by an experiment in which a FLAG-tagged BiP$^{V461F}$ (defective in substrate binding) was introduced into CHO-K1 cells alongside (untagged) wildtype BiP or (untagged) BiP$^{ADDA}$ and the effect of the co-expressed (untagged) BiP on the disposition of the interdomain linker of the FLAG-tagged BiP$^{V461F}$ mutant was probed by cleavage with SubA. In the absence of exogenous BiP, or in the presence of over-expressed wildtype BiP, all FLAG-tagged BiP$^{V461F}$-containing species were affected by SubA, indicating that in each species at least one protomer has an exposed, cleavable interdomain linker (*Figure 7D*, lanes 3 & 6). By contrast, oligomers assembled in the presence of the (untagged) BiP$^{ADDA}$ were strongly protected from SubA-mediated cleavage (*Figure 7D*, lane 9). In addition, the oligomers of FLAG-tagged BiP$^{V461F}$ formed in the presence of untagged BiP$^{ADDA}$ were maintained when ATP was added to the lysate (*Figure 7D*, lane 10). These observations point to BiP$^{ADDA}$, which is both resistant to cleavage by SubA and impaired in nucleotide-dependent allosteric transitions (*Figure 5—figure supplements 1, 2*) (and thus unable to release substrates upon exposure to ATP), engaging the interdomain linker of the FLAG-tagged BiP$^{V461F}$ in vivo and protecting it from digestion (*Figure 7—figure supplement 1B*).

## BiP's oligomeric status is subordinate to physiological conditions in the ER

Protein folding homeostasis in the ER can be manipulated experimentally by attenuating the supply of newly synthesized proteins through the use of translational inhibitors or by exposing cells to drugs that

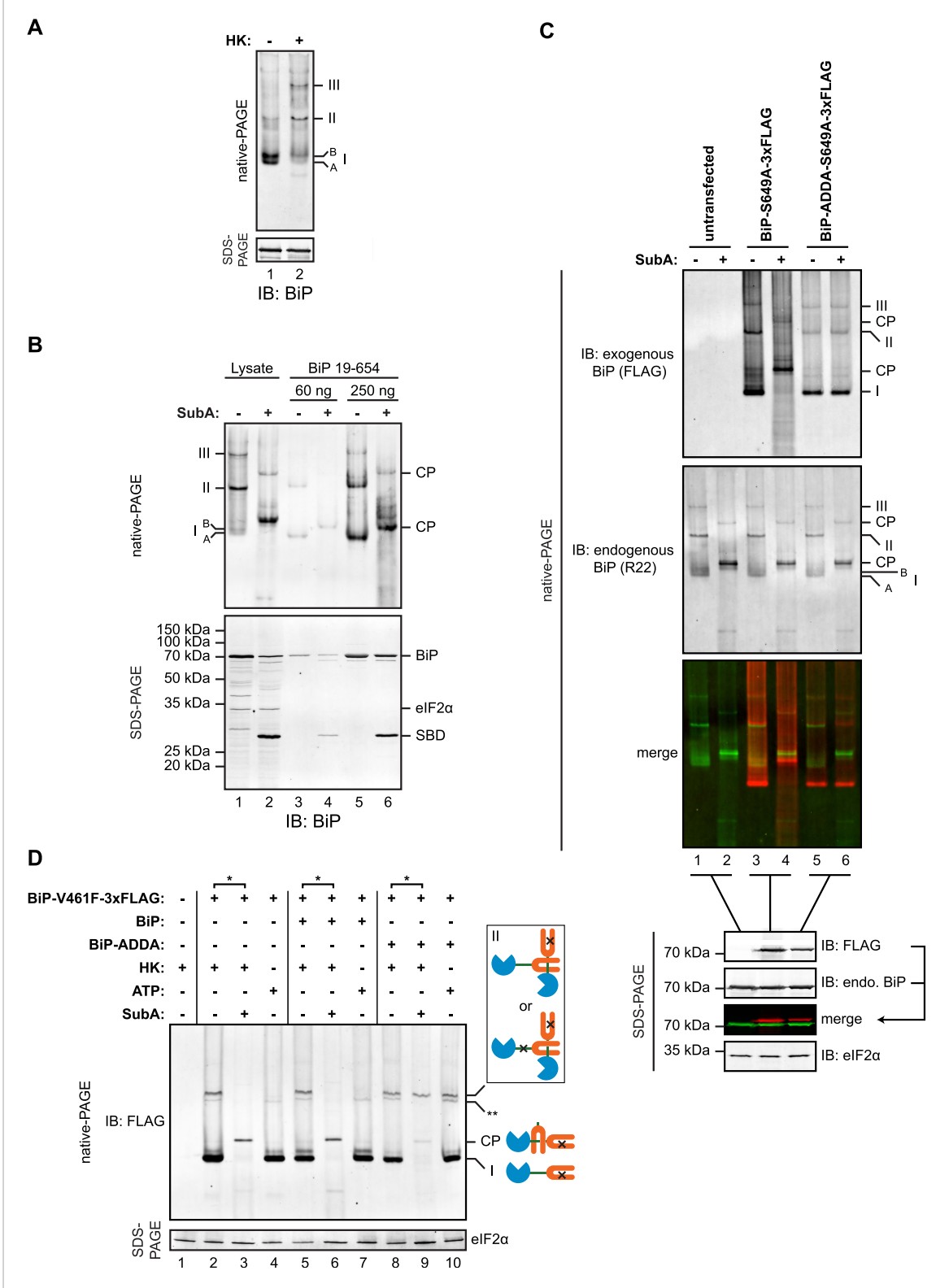

**Figure 7**. *Trans*-engagement of the linker by the substrate binding domain (SBD) as the basis for BiP oligomerization in vivo. (**A**) Immunoblot of endogenous BiP from lysates of CHO-K1 cells resolved by native-PAGE. Where indicated the lysate was depleted of ATP by incubation with hexokinase and glucose (HK). The major species observed on the native gel have been numbered I-III by order of descending mobility and the two forms of BiP detected in the higher mobility region of the gel ('A' and 'B') are indicated. Immunoblot of the same sample resolved by SDS-PAGE (lower panel) serves as
*Figure 7. continued on next page*

*Figure 7. Continued*

a loading control. (**B**) Immunoblot of endogenous BiP from CHO-K1 cells lysed in presence of hexokinase and glucose to deplete ATP (lysate) or mature hamster BiP (19–654) expressed in *Escherichia coli* and resolved on the same native gel. Where indicated the lysate and or pure BiP were exposed to SubA (5 ng/µl for 10 min at room temperature). The emergent cleavage products (CP) are indicated. Immunoblot of the same sample resolved by SDS-PAGE (lower panel). The free nucleotide binding domain (NBD) is undetected by this antiserum that recognizes the C-terminus of BiP. (**C**) Anti-FLAG and anti-BiP immunoblots of native gels of ATP-depleted lysates from CHO-K1 cells transfected with the indicated BiP constructs bearing an S649A mutation abolishing their reactivity with the anti-BiP R22 serum. The lysates were split and exposed to SubA as in 'B'. Anti-FLAG, anti-endogenous BiP (R22) and anti-eIF2α (a loading control) immunoblots of the same samples resolved by SDS-PAGE are presented in the panels below. Note that the FLAG-tagged exogenous BiP proteins are not recognized by the polyclonal anti-rodent BiP antiserum due to the S649A substitution. (**D**) Anti-FLAG immunoblot of ATP-depleted lysates from CHO-K1 cells transfected with the indicated BiP derivatives and resolved by native-PAGE. Where indicated, the two samples from the same lysate (*) were treated with or without SubA as in 'B'. ATP (3 mM) was added to lysates loaded in lanes 4, 7 and 10 (prepared without hexokinase and glucose) to dissociate oligomeric species. Anti-eIF2α immunoblot of the same samples resolved by SDS-PAGE is presented in the panel below as a loading control. To aid interpretation, a schema of the BiP species involved is provided to the side of the gel. The BiP NBD is colored blue its SBD orange and the interdomain linker green. The V461F mutation (in the SBD) and the ADDA mutation (in the interdomain linker) are indicated by crosses. Note that the BiP complexes represented by species II in lanes 9 and 10 are resistant to both, SubA and ATP. The identity of the band marked with ** is unknown. Note: To promote BiP oligomerization in vivo, the CHO-K1 cells in panels B-D were treated with thapsigargin (0.5 µM for 10 min before harvest, as will be explained in detail below).

The following figure supplement is available for figure 7:

**Figure supplement 1**. Defective homo-oligomerization of ADDA mutant BiP.

alter the protein-folding environment in the ER lumen. Exposure of cells to the protein synthesis inhibitor cycloheximide had a profound effect on the distribution of anti-BiP reactive species in the native gel. Both the defined oligomers and the generalized intra-lane background (the latter likely reflecting BiP interactions with unfolded client proteins) were diminished and replaced by a prominent high mobility species (labeled 'B' form, *Figure 8A*, lanes 1 and 2). This was evident in the native lysates and was accentuated following ATP depletion in vitro with hexokinase and glucose (*Figure 8A*, lanes 5 and 6). The accumulation of the 'B' form was blocked by addition of novobiocin, an inhibitor of ADP ribosylation, which also restored the oligomers in the cycloheximide-treated sample (*Figure 8A*, lane 7).

Elaboration of the 'B' form and depletion of oligomers was also observed by application of an orthogonal method to attenuate the burden of newly synthesized unfolded proteins in the ER: Activation of a modified eIF2α kinase, Fv2E-PERK, by a dimerising drug AP20187 [that sidesteps the normal induction of ER stress but inhibits translation initiation (*Lu et al., 2004*)], also led to accumulation of the 'B' form of BiP, apparently at the expense of oligomeric forms of BiP (*Figure 8B*, lane 5).

Kinetic analysis showed concordance between dissociation of the oligomers and appearance of the 'B' form (*Figure 8—figure supplement 1A*), which furthermore occurred on a timescale previously reported for accumulation of ADP ribosylated BiP (*Laitusis et al., 1999*) or AMPylated BiP (*Ham et al., 2014*) in cycloheximide-treated cells. Challenging cells with agents that perturb the protein folding environment in the ER and thereby increase BiP's clientele also led to diminution of oligomers, but without a commensurate increase in the 'B' form. This was evident in cells treated with the glycosylation inhibitor tunicamycin, the proline analog azetidine and the reducing agent dithiothreitol (DTT) and is consistent with competition between substrates and other BiP molecules for the chaperone's SBD (*Figure 8C,D* and *Figure 8—figure supplement 1B*).

The ER lumenal calcium-depleting agent, thapsigargin, a strong activator of the UPR (*Wong et al., 1993*), stood out as an important exception: Unlike the other manipulations, which activate the UPR by affecting the structure of ER client proteins, thapsigargin, an inhibitor of the smooth ER calcium reuptake (SERCA) pump that depletes ER calcium (*Sagara and Inesi, 1991*), promoted BiP oligomerization (*Figure 8C*, lane 5). This effect of thapsigargin was dominant to that of cycloheximide (which favors the 'B' form over oligomers, *Figure 8C*, lane 4) and to that of agents that directly perturb ER client protein structure: Tunicamycin (*Figure 8D*, lane 4), azetidine (*Figure 8D*, lane 6) or DTT (*Figure 8D*, lane 11).

## Physiological release of lumenal calcium promotes BiP oligomerization at the expense of unfolded protein substrate binding

The distribution of BiP into oligomeric forms followed rapidly upon depletion of lumenal calcium, being observed within 10 min of application of thapsigargin (*Figure 9A*, compare lanes 2 and 3), a

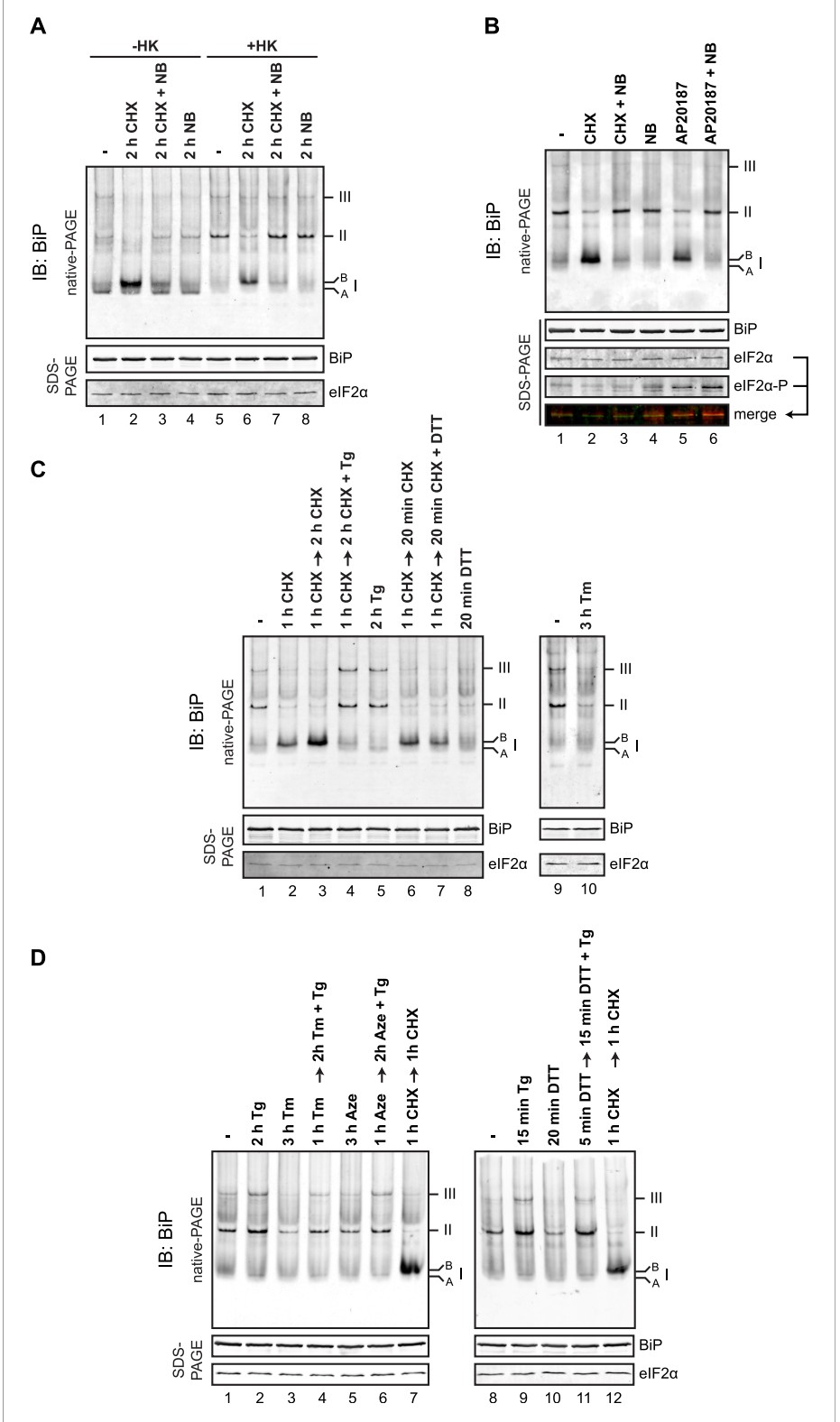

**Figure 8**. BiP oligomeric status responds to changes in ER unfolded protein load. (**A**) Immunoblot of endogenous BiP from CHO-K1 cell lysates resolved by native-PAGE. Where indicated the cells were exposed to cycloheximide (CHX, 100 μg/ml), novobiocin (NB, 0.5 mM) or both and the lysate was depleted of ATP by incubation with hexokinase and glucose (+HK). The major species observed on the native gel are numbered by order of descending mobility (I-III) and the 'B' form associated with cycloheximide and 'A' form present in untreated cells are noted.
*Figure 8. continued on next page*

*Figure 8. Continued*

Immunoblot of the same sample resolved by SDS-PAGE (lower panel) reports on total BiP loaded and on eIF2α as a loading control. (**B**) As in 'A'. ATP-depleted lysates of CHO-K1 cells harboring a stable Fv2E-PERK transgene encoding a dimerizer drug (AP20187)-inducible form of the eIF2α-kinase PERK. Where indicated the cells were exposed for 2 hr to AP20187 (15 nM) to activate Fv2E-PERK, in the presence or absence of novobiocin. The lower panels report on the levels of BiP, total eIF2α and phosphorylated eIF2α in the lysates. (**C**, **D**) As in 'A'. Where indicated, cells were treated alone or sequentially with cycloheximide (CHX, 100 µg/ml) and the lumenal calcium depleting agent thapsigargin (Tg, 0.5 µM), the reducing agent DTT (1 mM) that interferes with disulfide bond formation, the glycosylation inhibitor tunicamycin (Tm, 2.5 µg/ml) or the protein misfolding-inducing proline analog, azetidine (Aze, 4 mM) in the indicated order and time. ATP was depleted from all samples during cell lysis.

The following figure supplement is available for figure 8:

**Figure supplement 1**. Time-course of evolution of the cycloheximide-dependent 'B' form of BiP.

---

point at which lumenal calcium levels [assessed by the fluorescent ER calcium probe, D1ER cameleon (*Palmer et al., 2004*)] declined markedly (*Figure 9B*). Oligomerization was sustained throughout the period of lumenal calcium depletion and was noted well before the thapsigargin-mediated attenuation in the levels of the 'B' form of BiP (*Figure 9A*, lane 6). The latter observation suggests that the well documented mobilization of the modified (presumably ADP-ribosylated/AMPylated) 'B' form of BiP by ER calcium depletion (*Laitusis et al., 1999*; *Ham et al., 2014*) occurs on a timescale that is too long to account for the rapid increase in BiP oligomers upon ER calcium depletion and that a pool of BiP, distinct from the 'B' form, is tapped for oligomerization early upon calcium release. Similar observations were made in CHO-K1 cells exposed to the calcium ionophore A23187, pointing to the role of lumenal calcium fluctuations (as opposed to some other effect of thapsigargin) in the re-distribution of BiP amongst its various forms (*Figure 9C,D*).

To determine if physiological levels of calcium mobilization also resulted in changes in BiP's oligomeric status we exploited the presence of purinergic receptors on the surface of CHO-K1 cells to promote ER calcium depletion by their engagement with a physiological ligand, extracellular ATP. As expected, application of ATP to the growth medium led to a rapid and transient depletion of lumenal calcium stores (*Hayashi and Su, 2007*) (*Figure 10A*). This was correlated with a rapid and transient increase in BiP oligomers, observed over a similar time frame (*Figure 10B*). Extracellular ATP was linked to BiP oligomerization through a canonical signaling cascade, as it was attenuated by simultaneous application of the phospholipase C blocking drug U73122 (*Figure 10C*, lane 4). Phospholipase C inhibition had no effect on the downstream-acting thapsigargin's ability to promote BiP oligomerization (*Figure 10C*, lane 6), pointing to the specificity of U73122 in interfering with BiP oligomerization by attenuating signaling downstream of the purinergic receptor.

As lumenal calcium depletion-mediated oligomerization does not appear to draw, initially, on the inactive 'B' form of BiP, we sought to examine the alternative, whereby thapsigargin-mediated calcium depletion might have correlated with dissociation of BiP from its unfolded clients to form oligomers. ER calcium depletion led to diminished recovery of BiP in complex with a misfolded truncated ER client protein, the product of the null Hong Kong allele of α1-antitrypsin (*Liu et al., 1997*), introduced by transfection into CHO-K1 cells (*Figure 11A*). Similar observations were made in regard to the complex between BiP and immunoglobulin heavy chain (IgHC, expressed in the absence of the light chain); it too was observed to partially dissociate in thapsigargin-treated cells (*Figure 11B*).

To test the impact of physiological calcium depletion from the ER on the abundance of complexes between endogenous BiP and an endogenous client, we turned to pancreatic AR42j cells, which differentiate to amylase-producing acinar cells by exposure to dexamethasone. Consistent with increased synthesis of secretory proteins, fewer BiP oligomers were observed in differentiated cells whereas stimulation with cholecystokinin, which engages a cell surface receptor linked to ER calcium depletion (*Zhao et al., 1990*; *Pinnock et al., 1994*), enhanced oligomerization (*Figure 11—figure supplement 1*). Cholecystokinin also promoted dissociation of amylase, a major secretory product of pancreatic acinar cells, from BiP (*Figure 11C*). Reciprocally, BiP associated with

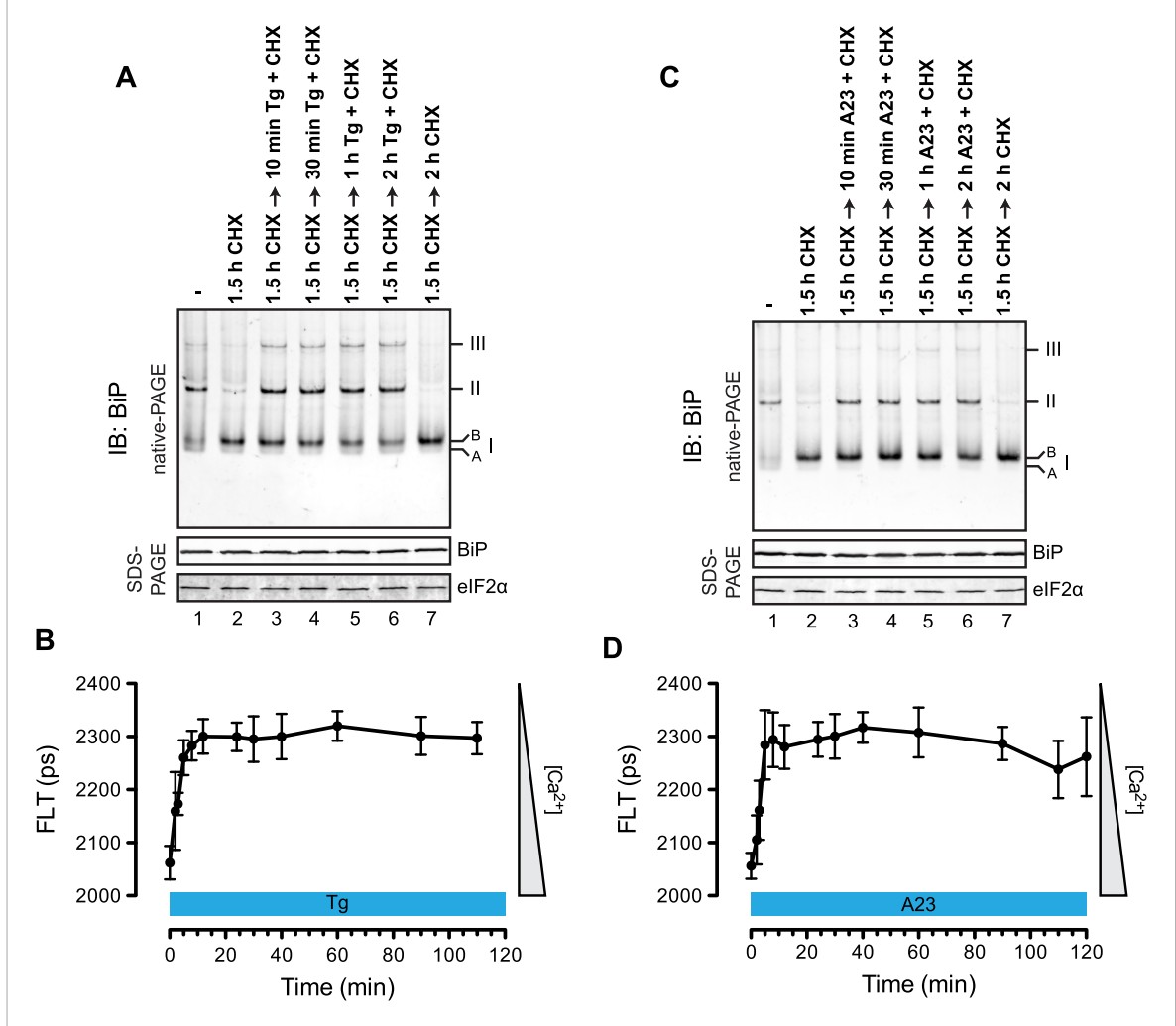

**Figure 9**. Rapid BiP oligomerization in response to ER calcium depletion. (**A**) Immunoblot of endogenous BiP from ATP-depleted lysates of CHO-K1 cells resolved by native-PAGE. Where indicated the cells were exposed to cycloheximide (CHX, 100 μg/ml) followed by thapsigargin (Tg, 0.5 μM). The major species observed on the native gel have been numbered by order of descending mobility (I-III) and the 'B' and 'A' forms noted. Immunoblot of the same samples resolved by SDS-PAGE (lower panel) reports on total BiP loaded and on eIF2α as a loading control. (**B**) Plot of time-dependent change of the donor fluorescence lifetime (FLT) of ER localized D1ER cameleon in CHO-K1 cells treated for 1.5 hr with CHX followed by exposure to Tg as in 'A'. The increase in donor fluorescence lifetime, a consequence of loss of calcium-dependent intra-molecular FRET in cameleon, reports on the kinetics of ER calcium depletion. Bars: mean ± SEM. (**C**) As in 'A', substituting the calcium ionophore A23187 (A23, 10 μM) for thapsigargin. (**D**) As in 'B' substituting the calcium ionophore A23187 (A23, 10 μM) for thapsigargin.

amylase was also diminished by cholecystokinin (*Figure 11D*). The BiP:amylase complex that was diminished by calcium depletion in vivo was also sensitive to ATP in vitro (*Figure 11D*, lane 4), confirming that it represented engagement of a conventional client by BiP. These findings confirmed that ER lumenal calcium depletion, induced by a physiological ligand that promotes BiP oligomerization, also disfavored interaction of endogenous BiP with an endogenous client. Together, these observations establish a negative correlation between BiP oligomerization and client binding.

## Discussion

BiP oligomers were observed here to form by engagement of the hydrophobic interdomain linker of one BiP protomer as a typical substrate of an adjacent BiP protomer. This process, revealed by monitoring the accessibility of BiP's interdomain linker to a sequence-specific protease, SubA, and by site-specific crosslinking, is conspicuous in preparations of pure proteins in vitro, and in living cells. BiP

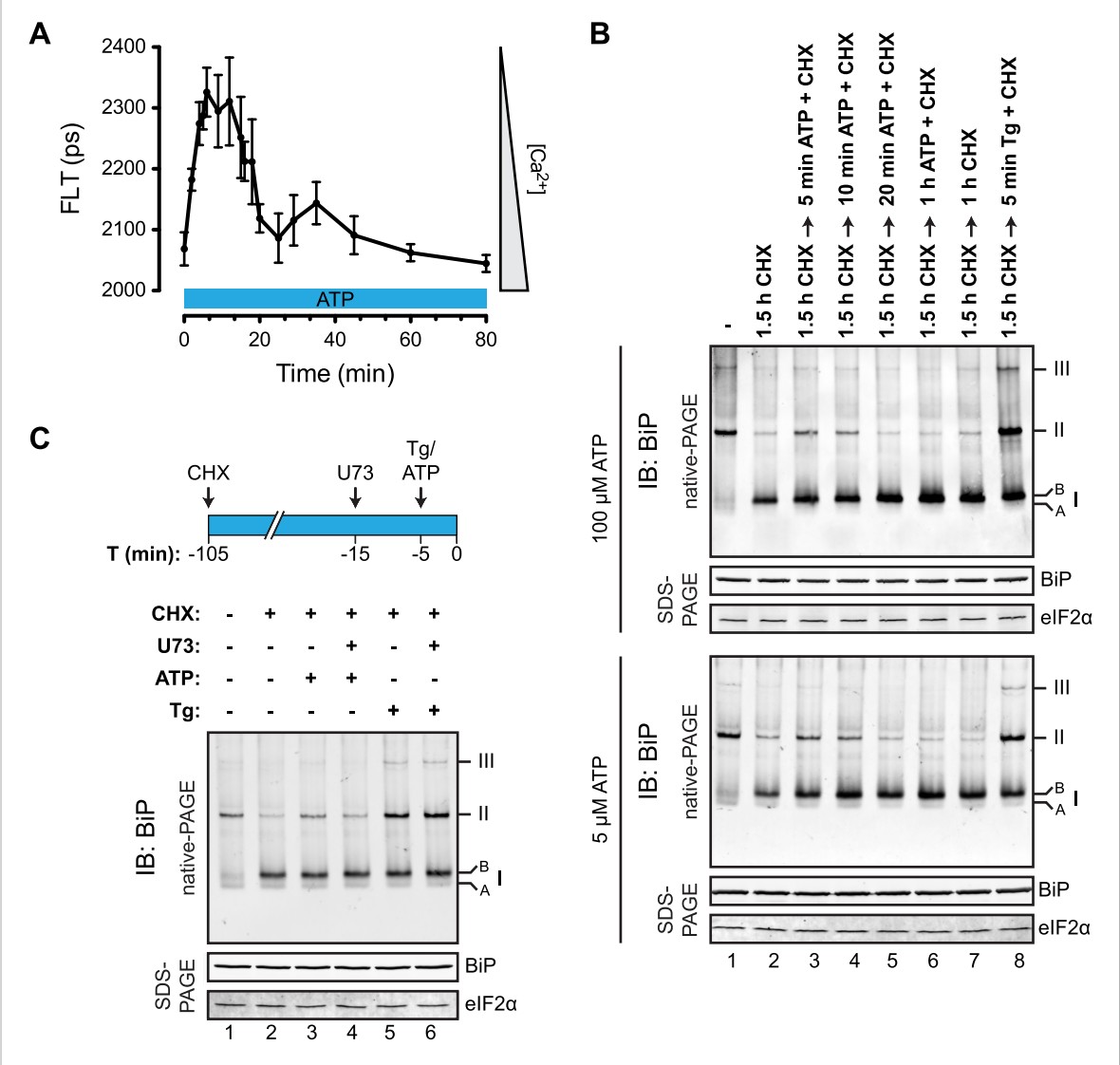

**Figure 10**. Purinergic receptor engagement promotes BiP oligomerization at physiological levels of ER calcium depletion. (**A**) Plot of time-dependent change if the donor fluorescence lifetime of ER localized D1ER cameleon in CHO-K1 cells treated for 1.5 hr with cycloheximide (100 μg/ml) followed by exposure to extracellular ATP (100 μM), a ligand of endogenous purinergic receptors. The increase in donor fluorescence lifetime, a consequence of loss of calcium-dependent intra-molecular FRET in D1ER cameleon, reports on the kinetics of ER calcium depletion, which is notably transient. Bars: mean ± SEM. (**B**) Immunoblot of endogenous BiP from ATP-depleted lysates of CHO-K1 cells resolved by native-PAGE. The major species observed on the native gel have been numbered by order of descending mobility (I-III) and the 'A' and 'B' forms are indicated. Where indicated the cells were exposed to cycloheximide (CHX, 100 μg/ml) followed by ATP. Immunoblots of the same samples resolved by SDS-PAGE are shown below the native gels and report on total BiP loaded and on eIF2α as a loading control. (**C**) Native-PAGE immunoblot of endogenous BiP. As indicated in the schema above, cells were treated sequentially with CHX (100 μg/ml), ATP (100 μM), thapsigargin (Tg, 0.5 μM) or the phospholipase C inhibitor U73122 (U73, 6 μM) in the indicated order. DMSO was used instead of U73 as a vehicle control. Immunoblot of the same samples resolved by SDS-PAGE (lower panel) reports on total BiP loaded and on eIF2α as a loading control.

oligomerization is rapidly tuned to the physiological state of the cell, and given that oligomerization inactivates participating BiP protomers, the oligomeric form is a relatively inactive storage form of the chaperone that is accessible on a short time scale. Our findings are consistent with a model whereby competition between oligomerization and substrate binding buffers the ER against rapid fluctuations in unfolded protein load in both directions.

The interdomain linker of Hsp70 chaperones consists of a short stretch of aliphatic residues, well suited to engage the SBD (*Flynn et al., 1989*; *Blond-Elguindi et al., 1993*). A crystal form of bacterial

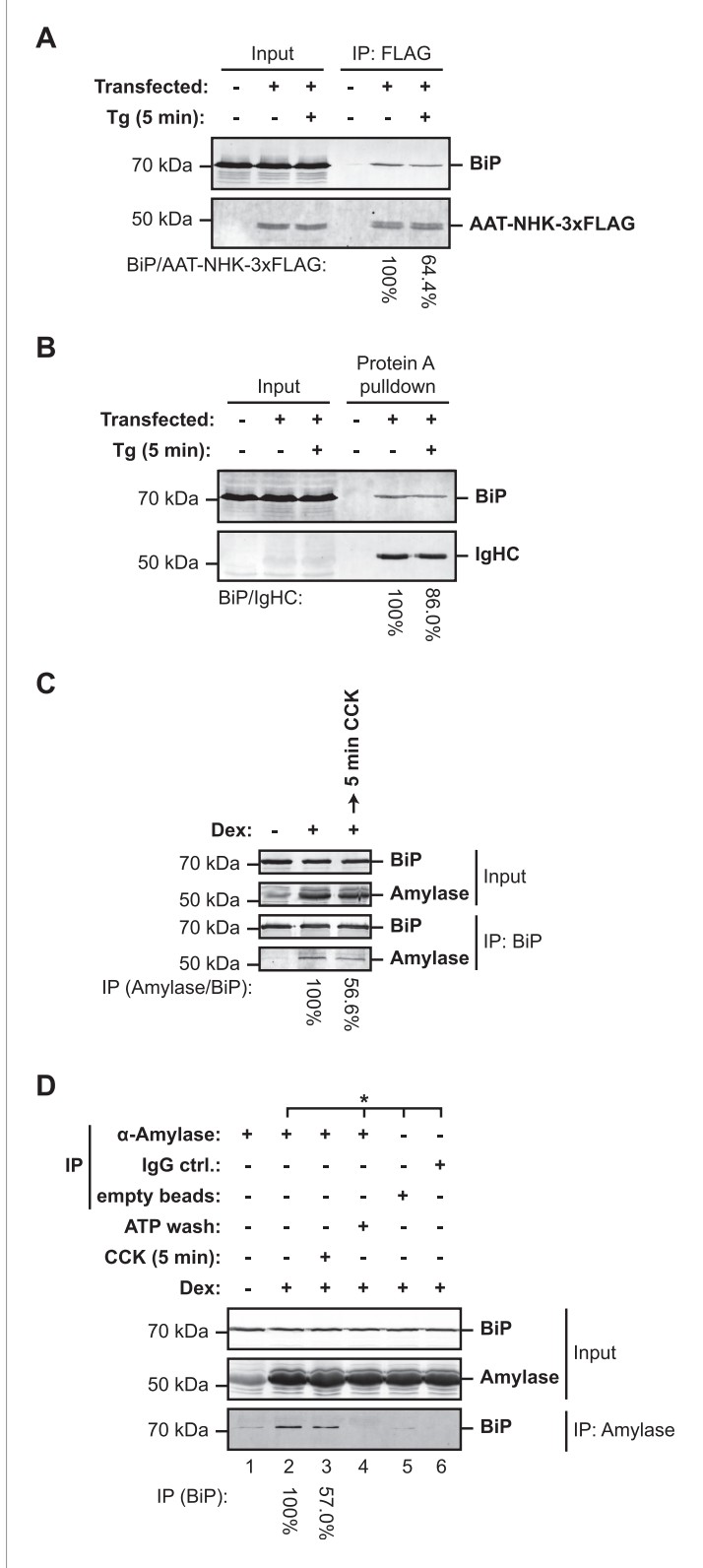

**Figure 11.** Lumenal calcium depletion-mediated dissociation of BiP from its client proteins. (**A**) SDS-PAGE and immunoblot of BiP and FLAG-tagged misfolded null Hong Kong variant of α1-antitrypsin (AAT-NHK-3xFLAG) in ATP-depleted lysates of transfected CHO-K1 cells (Input) or recovered in an anti-FLAG immunoprecipitation. Where indicated the cells were exposed to thapsigargin (Tg, 1 µM) for 5 min before lysis. The recovery of BiP, normalized to

*Figure 11. continued on next page*

*Figure 11. Continued*

the α1-antitrypsin signal in the immunoprecipitation, is noted below each lane and is set to 100% in the untreated sample. Shown is a representative experiment reproduced three times with similar, Tg-mediated disruption of the BiP-AAT complex. (**B**) SDS-PAGE and immunoblot of BiP and immunoglobulin heavy chain (IgHC) in ATP-depleted lysates of transfected CHO-K1 cells (Input) or recovered in complex with *S. aureus* protein A. Where indicated the cells were exposed to Tg (1 μM) for 5 min before cell lysis. The recovery of BiP, normalized to the immunoglobulin heavy chain signal in the protein A complex, is noted below each lane and is set to 100% in the untreated sample. Shown is a representative experiment reproduced three times with similar, Tg-mediated disruption of the BiP-IgHC complex. (**C**) SDS-PAGE and immunoblot of endogenous BiP and amylase in ATP-depleted lysates of AR42j cells (Input) or following immunoprecipitation of BiP from the same (IP). Where indicated the cells were differentiated into an acinar lineage with dexamethasone (Dex, 100 nM) for 48 hr followed by exposure to cholecystokinin (CCK, 4 μM) for 5 min. The recovery of amylase, normalized to the BiP signal in the immunoprecipitation is noted below each lane and is set to 100% in the untreated sample. Shown is a representative experiment reproduced three times with similar, CCK-mediated disruption of the BiP-amylase complex. (**D**) SDS-PAGE and immunoblot of endogenous BiP and amylase from ATP-depleted lysates of AR42j cells (Input) and recovered in an anti-amylase immunoprecipitation (IP), or two mock IPs. The cells were differentiated into the acinar lineage by dexamethasone (Dex) as in 'C' and exposed to cholecystokinin (4 μM), as indicated. The recovery of BiP, normalized to the amylase signal in the immunoprecipitation, is noted below each lane and is set to 100% in the untreated sample. Samples from the same lysate are marked with an asterisk (*). The immunoprecipitated sample in lane 4 was subjected to an ATP elution step, to confirm that BiP associated with amylase through a conventional substrate-binding mechanism. Shown is a representative experiment reproduced three times with similar, CCK-mediated disruption of the amylase–BiP complex.

The following figure supplement is available for figure 11:

**Figure supplement 1**. Cholecystokinin-mediated lumenal calcium depletion promotes BiP oligomerization in AR42j cells.

DnaK with cross-protomer binding of the interdomain linker in the SBD shows the engagement of two highly conserved aliphatic side chains of the linker in hydrophobic sites of the SBD (*Chang et al., 2008*). Given the role of these very same side chains in allosteric coupling between the NBD and SBD, this mode of BiP oligomerization results in removal of the BiP protomers involved from the pool of active chaperones. The proposed architecture of the oligomers (*Figure 1C*) fits well with our finding of a single peptide binding site available in BiP oligomers formed in vitro, regardless of their size (*Figure 2F*). The model fits the observation that oligomeric forms of Hsp70 are holdases rather than foldases (*Blond-Elguindi et al., 1993*; *Thompson et al., 2012*), as the only protomer capable of engaging substrate (protomer A, *Figure 1C*) is allosterically uncoupled by the engagement of its interdomain linker in the SBD of the adjacent protomer. The model also explains the observation that DnaK oligomers formed in vitro are subject to regulation by the nucleotide exchange factor, GrpE (*Thompson et al., 2012*), as the last protomer (protomer C, *Figure 1C*) is predicted to retain the capacity to respond allosterically to ATP binding to its NDB, initiating oligomer disassembly.

Our in vitro experiments provide a crude guide to the stability of BiP oligomers. In the absence of ATP, oligomers appear to be remarkably stable with a dissociation $K_{1/2max}$ in the lower micromolar range. In vivo, oligomer assembly and disassembly is likely to be influenced by co-chaperones of the DnaJ family that enhance ATP hydrolysis to promote the high affinity state and exchange factors, such as GRP170 and SIL1, that convert BiP to the low affinity state by exchanging ADP for ATP. These co-factors are likely to have an important role in the kinetics of the inter-conversion of BiP monomers to oligomers and in the competing processes of substrate binding and release. Nonetheless, the high concentration of BiP in the ER (>200 μM, [*Ghaemmaghami et al., 2003*]) and the inherent affinity of BiP protomers for one another, are consistent with a model whereby oligomerization and substrate binding compete with each other on a level playing field.

Exploring the relationship between unfolded protein load and the oligomeric state of Hsp70 chaperones is confounded by powerful feedback that dominates the unfolded protein responses and couples chaperone production to the levels of unfolded protein stress. To explicitly cope with this methodological limitation, we have confined the analysis to the early stages of the perturbations that increase the unfolded protein load in the ER and employed manipulations, such as reduction of

disulfide bonds by DTT, that are effective even in the absence of ongoing protein synthesis. Our observations paint a coherent picture whereby enhanced production of unfolded proteins rapidly draws on the pool of inactive, oligomeric BiP to service its growing clientele. Reports of enhanced presence of oligomers in longer-term stressed cells may have reflected the partitioning of excess chaperone produced during the activated UPR (or its cytoplasmic counterpart, the heat shock response) but made redundant following resolution of the insult (*Freiden et al., 1992*; *Thompson et al., 2012*). Furthermore, activation of parallel chaperones, such as the oxidative stress induced Hsp33 of bacteria, may account for regulated redundancy of the Hsp70/DnaK system, favoring oligomerization through linker engagement in the course of certain stress responses (*Winter and Jakob, 2004*).

The ER is unusual in possessing alternative mechanisms for inactivating its Hsp70 protein, BiP, by covalent modification (ADP ribosylation) (*Freiden et al., 1992*; *Laitusis et al., 1999*; *Chambers et al., 2012*) and AMPylation (*Ham et al., 2014*). The modified form of BiP appears to equilibrate with the active pool more slowly than the oligomers. Together, our observations suggest a hierarchical post-translational mechanism for maintaining balance between chaperones and their clients in the highly dynamic ER (*Figure 12*). The first line of buffering consists of oligomer formation, as the balance between chaperones and clients tips in favor of the former, and oligomer dissolution, as the balance tips the other way. Conversion of BiP to the modified, inactive form proceeds on a longer time scale. Its precise relationship to oligomerization is unclear, as both the unemployed monomer and inactive oligomer, could, in theory, serve as substrates for the modification(s). Either way it seems likely that the physiological impact of both oligomerization and modification is to enable the cell to avoid the cost of over-chaperoning (*Dorner et al., 1992*; *Feder et al., 1992*) without having to degrade surplus BiP as stress is resolved and to provide cells with an inactive reserve of BiP to draw upon as stress mounts, without resorting to activation of the translational and transcriptional arms of the UPR.

Our analysis has been confined to the ER, where the balance between chaperones and clients fluctuates widely according to the physiological state of the cell and is relatively easy to manipulate pharmacologically. But oligomerization can serve as an intrinsic mechanism to inactivate excess Hsp70 in other compartments of the eukaryotic cell and indeed across phyla. In that sense oligomerization would appear to be a common primitive strand of the post-translational UPR, recruiting and inactivating Hsp70 chaperones according to need. This simple, mass action-based mechanism for regulating Hsp70 function may exist alongside other mechanisms, such as oxidation-induced domain unfolding, which may promote alternative modes of oligomerization (*Winter et al., 2005*). Such alternative modes may account for the heterogeneity of DnaK oligomers observed by single particle electron microscopy (*Thompson et al., 2012*).

The inverse correlation between unfolded protein burden and BiP oligomerization appears to break down in the context of ER calcium depletion, which is both a strong inducer of the UPR and promotes rapid oligomerization of BiP. This apparent paradox needs to be considered in the context of observations that BiP oligomerization upon calcium depletion also correlates with less substrate binding (a feature first noted many years ago [*Suzuki et al., 1991*] and confirmed here) and with the observation that

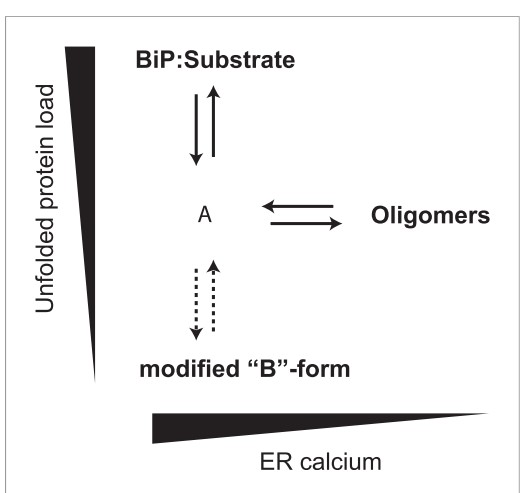

**Figure 12**. Schema of the proposed relationships between the different forms of BiP. The monomer ('A' form) at the center of the schema binds clients and is shifted to BiP:Substrate complexes by mounting unfolded protein load. With diminishing clients it partitions to inactive oligomers, with whom it exists in equilibrium. This equilibrium is influenced by ER calcium, whose depletion favors the oligomeric form. The 'A' form is also in equilibrium with modified BiP ('B' form, likely ADP-ribosylated or AMPylated BiP) and this equilibrium too is influenced by unfolded protein burden but on a slower time scale than oligomer formation and disassembly.

calcium depletion-mediated oligomerization slowly draws on the modified, inactive pool of BiP. From BiP's perspective, oligomerization appears to be the primary event triggered by ER calcium efflux. This depletes the pool of active chaperones and disrupts, by mass action, chaperone-client interactions, releasing clients that would otherwise have remained bound to BiP. The ensuing, slower loss of the modified form of BiP is consistent with equilibrium between the modified form and the monomer and its latency is consistent with the need for enzymes to remove the modifications.

We do not understand how calcium depletion shifts the equilibrium in favor of the oligomer. In vitro, there is no measureable effect of calcium on BiP oligomerization, in either the ATP state or ADP state or in the presence of the J-factor ERdj3 or the exchange factor GRP170 (not shown), yet in vivo, ER calcium depletion promotes such oligomerization. This discrepancy could be resolved by an ER-localized J-factor whose affinity for BiP as a substrate is increased selectively as $(Ca^{+2})_{ER}$ declines. Given that the affinity of Hsp70 proteins for their substrates is governed by J-protein driven non-equilibrium mechanisms (*De Los Rios and Barducci, 2014*), such a scenario could account for BiP favoring itself as a client (oligomerizing) over other clients, as $(Ca^{+2})_{ER}$ declines. Regardless of mechanism, one consequence of $(Ca^{+2})_{ER}$ depletion-mediated BiP oligomerization would be to also shift the equilibrium between unfolded proteins and active chaperones in favor of the former. Furthermore, given the 1:1 stoichiometry of the repressive complex formed between BiP and the lumenal, stress-sensing domains of the UPR transducers PERK and IRE1 (*Bertolotti et al., 2000*), it seems likely that the monomer serves as a UPR repressor. Thus, $(Ca^{+2})_{ER}$ depletion-mediated oligomerization of BiP stands to contribute to UPR activation by increasing the burden of free unfolded proteins and by lowering the availability of monomeric BiP to repress the stress transducers.

Calcium depletion-mediated oligomerization of BiP is observed in physiological circumstances, as a consequence of engagement of cell surface receptors by activating ligands that couple to store release channels. We do not know if oligomerization of BiP and its downstream consequences are a feature that promotes fitness (i.e. a feature that has been positively selected during eukaryote evolution) or simply a reflection of limitations in the functionality of an organelle that experiences fluctuations in the concentration of calcium, imposed by other exigencies, such as the benefits of signaling in the cytosol by the released calcium. However, it is tempting to speculate that the transient excursions of UPR signaling imposed by BiP oligomerization may have some benefit. In secretory cells, such a scheme would couple signals that promote granule secretion to transient translational repression (attendant upon PERK activation [*Harding et al., 1999*]). This may have short-term benefits in terms of diverting resources away from translation to the bio-energetically costly process of secretion. Transient activation of UPR transducers also results in elaboration of downstream effectors. These transcription factors, XBP1, ATF6 and ATF4, adapt cells to their secretory mission by promoting the latent process of ER biogenesis (*Sriburi et al., 2004*; *Lee et al., 2005*), and by expanding the capacity to translate mRNAs (*Harding et al., 2003*). It is thus tempting to speculate that calcium depletion-mediated BiP oligomerization contributes to transient activation of the UPR observed in secretory cells responding to secretagogues (*Kubisch and Logsdon, 2007*) and thereby promotes long-term coupling of the capacity to produce and process precursors of secretory proteins to the short-term demand for the secreted end product.

## Materials and methods

### Plasmid construction

*Supplementary file 1* lists the plasmids used, their lab names, description and notes their first appearance in the figures and the corresponding label, and provides a past reference, where available.

### Protein purification

Wildtype and mutant versions of N-terminally hexahistidine- (His6-) tagged hamster BiP (see *Supplementary file 1* and 'Plasmid construction') were encoded on pQE10 plasmids (Qiagen UK, Manchester, United Kingdom) and expressed in M15 *E. coli* cells (Qiagen) as described previously (*Chambers et al., 2012*). In brief, bacterial cultures were grown at 37°C in LB medium containing 100 µg/ml ampicillin and 50 µg/ml kanamycin to an optical density ($OD_{600\ nm}$) of 0.8 and expression was induced with 1 mM isopropylthio β-D-1-galactopyranoside (IPTG). After incubation for 6 hr at

37°C the cells were sedimented by centrifugation and pellets were lysed using a high-pressure homogenizer (EmulsiFlex-C3; Avestin) in buffer A [50 mM Tris–HCl pH 7.5, 500 mM NaCl, 1 mM $MgCl_2$, 0.2% (vol/vol) Triton X-100, 10% (vol/vol) glycerol, 20 mM imidazole] supplemented with protease inhibitors (2 mM phenylmethylsulphonyl fluoride (PMSF), 4 µg/ml pepstatin, 4 µg/ml leupeptin, 8 µg/ml aprotinin) and 0.1 mg/ml DNaseI. The lysates were cleared by centrifugation for 30 min at 25,000 $g$ and incubated with 1 ml Ni-NTA agarose (Quiagen) per 1 L of expression culture for 2 hr at 4°C. The beads were washed five times with 20 bed volumes of buffer A sequentially supplemented with 30 mM imidazole, 1% (vol/vol) Triton X-100, 1 M NaCl, 5 mM $Mg^{2+}$-ATP or 0.5 M Tris–HCl pH 7.5. Bound BiP protein was eluted in buffer B (50 mM Tris–HCl pH 7.5, 500 mM NaCl, 1 mM $MgCl_2$, 10% (vol/vol) glycerol, 250 mM imidazole) and dialyzed against HKM buffer (50 mM HEPES-KOH pH 7.4, 150 mM KCl, 10 mM $MgCl_2$). The purified BiP proteins were concentrated using centrifugal filters (Amicon Ultra, 30 kDa MWCO; Merck Millipore), snap-frozen in liquid nitrogen and stored at −80°C.

Mature hamster BiP (19–654) was purified via an N-terminal His6-Smt3 tag, which can be removed by cleavage with the Smt3-specific protease Ulp1. For that the hamster BiP-encoding sequence was inserted downstream of a His6-Smt3-encoding sequence into pCA528 (*Andreasson et al., 2008*) resulting in a His6-Smt3-BiP fusion construct (see *Supplementary file 1* and 'Plasmid construction'). The plasmids were transformed into the BL21(DE3) *E. coli* strain and cells were grown in LB medium containing 50 µg/ml kanamycin at 37°C to $OD_{600\ nm}$ 0.8. Production of the fusion protein was induced with 1 mM IPTG for 6 hr at 18°C. The cells were then harvested, lysed and His6-Smt3-BiP was affinity purified with Ni-NTA agarose beads as described above. The eluted protein was supplemented with 2 mM DTT and cleaved with Ulp1 (12 µg per mg of eluted protein) for 16 hr at 4°C during dialysis against HKM buffer containing 2 mM β-mercaptoethanol. The sample was then supplemented with 1 mM ATP and passed over a Superdex 200 10/300 GL gel filtration column (GE Healthcare) in buffer C (50 mM HEPES-KOH pH 7.4, 150 mM KCl, 20 mM imidazole, 10 mM $MgCl_2$). A 1 ml reverse Ni-affinity chromatography column (HisTrap HP; GE Healthcare) was connected in series with the gel filtration column to retain uncleaved full-length protein and the His6-Smt3 cleavage product. The BiP-containing fractions were pooled, diluted in buffer D (20 mM HEPES-KOH pH 7.4, 25 mM KCl) to reduce the ionic strength and loaded onto a Mono Q 5/50 GL anion exchange chromatography column (GE Healthcare). Bound protein was eluted with a linear salt gradient from 0 to 1 M NaCl in buffer D. The BiP-enriched eluate fractions were pooled and buffer was exchanged against HKM using a Centri•Pure P25 desalting column (emp BIOTECH). Purified authentic BiP was concentrated and stored as described above. Labelling of purified $BiP^{V461F;E652C}$ with the lucifer yellow iodacetimide fluorescent probe (Invitrogen) was performed according to the manufacturer's instructions.

## Analytical SEC

Purified BiP proteins were diluted from concentrated stocks to 50 µM in HKM buffer (50 mM HEPES-KOH pH 7.4, 150 mM KCl, 10 mM $MgCl_2$) and incubated (as described in the figure legends) in a final volume of 20 µl at room temperature for 16 hr before the experiments. For binding experiments the peptides (1 mM) or lucifer yellow (LY)-labeled probes (1 µM) were present during the 16 hr incubation period. Where indicated the samples were treated with ATP and SubA for 20 min before the SEC run. 10 µl of each sample were injected onto a SEC-3 HPLC column (300 Å pore size; Agilent Technologies) equilibrated with HKM and runs were performed at a constant flow rate of 0.3 ml/min at ambient temperature. Peptide bond absorbance traces at 230 nm ($A_{230\ nm}$) and LY fluorescence emission signals at 525 nm (excitation at 430 nm) were recorded and plotted against elution time. For quantification of incorporation of fluorescently labeled substrate peptide per BiP mass Gaussian curves were fitted to absorbance and fluorescence traces with the program MagicPlot Student 2.5.1 (Magicplot Systems, LLC) to determine the area under the peaks. The fluorescent signal was then normalized to the protein mass for each peak ($R^{FLU/A230\ nm}$) and expressed relative to peak I. For size reference a gel filtration standard (BioRad cat. # 151–1901) was applied and the elution peaks of Thyroglobulin (670 kDa), γ-globulin (158 kDa), Ovalbumin (44 kDa) and Myoglobulin (17 kDa) are indicated.

Multiangle light scattering (SEC-MALS) measurement of BiP monomers and oligomeric complexes was performed immediately following SEC by inline detection of static light scattering (DAWN 8+; Wyatt Technology), differential refractive index (Optilab T-rEX; Wyatt Technology), and UV absorbance at 280 nm (1260 UV; Agilent Technologies). A 100 µl sample of purified wildtype BiP

(90 µM) was passed through a Superdex 200 Increase 10/300 gel filtration column (GE Healthcare) equilibrated in HKM at a flow rate of 0.5 ml/min at room temperature. Molar masses were calculated using ASTRA 6 (Wyatt Technology).

## Malachite green ATPase assay

MG reaction solution was prepared freshly before the experiment by mixing stock solutions of MG dye (0.111% [wt/vol] malachaite green, 6 N sulphuric acid), 7.5% (wt/vol) ammonium molybdate and 11% (vol/vol) Tween 20 in a 10:2.5:0.2 ratio. Samples of purified BiP proteins (5 µM) were incubated in HKM with 3 mM ATP in a final volume of 20 µl for 60 min at 30°C. 15 µl of each sample were then diluted with 135 µl water on a 96-well plate, mixed with 50 µl MG reaction solution and incubated for 2 min at room temperature. 20 µl of 34% (wt/vol) sodium citrate were added to quench the reactions and after incubation for further 30 min the absorbance was measured at 612 nm with a Tecan Infinite F500 plate reader.

## Glutaraldehyde crosslinking

Chemical crosslinking with glutaraldehyde was adapted from (*Schlossman et al., 1984*; *Gao et al., 1996*). Purified BiP proteins (≥10 µM) were incubated with or without ATP or BiP-binding peptide (HTFPAVL) in HKM buffer at 25°C as described. Crosslinking was performed by diluting the samples to 0.18 µM final protein concentration in HKM buffer containing 0.05% (vol/vol) glutaraldehyde and short incubation for 2 min at 25°C. The reaction was quenched on ice with 80 mM sodium borohydride (NaBH$_4$) for 5 min followed by addition of 200 mM Tris–HCl pH 8 and further incubation for 5 min. The proteins were then precipitated with 20% (vol/vol) trichloroacetic acid (TCA) in presence of 0.2% (vol/vol) Triton X-100 for 16 hr on ice, washed with acetone and solubilized in alkaline SDS sample buffer. The proteins were separated on 7% sodium dodecyl sulfate (SDS) polyacrylamide-gels and stained with Coomassie. For dissociation experiments BiP was diluted from a 50 µM stock solution to 0.18 µM and incubated at 25°C for the indicated times before addition of glutaraldehyde, as described above.

## Site-specific crosslinking

Site-specific incorporation of the photoactivatable crosslinking amino acid *p*-benzoyl-$_L$-phenylalanine (pBpa) into C-terminally His6-tagged BiP proteins was performed using the amber codon suppression system developed in the Schultz lab (*Chin et al., 2002*). C-terminally His6-tagged BiP expression plasmids with site-specific amber (TAG) codons were constructed in the kanamycin-resistant pET30a vector (*Supplementary file 1*) and were co-transformed with pSup-BpaRS-6TRN (*Ryu and Schultz, 2006*), a plasmid encoding a mutant tRNA synthetase and amber suppressor tRNAs into BL21(DE3) *E. coli* cells and cultures from single clones were grown at 37°C in LB medium containing 50 µg/ml kanamycin, and 25 µg/ml chloramphenicol to an OD$_{600\,nm}$ of ~1.0. The cultures were then shifted to 30°C and pBpa (BACHEM cat. # F-2800) was added to a final concentration of 2 mM, 10 min prior induction with 0.5 mM IPTG for 4 hr at 37°C.

For small-scale purification of pBpa-derivatized BiP proteins (*Figure 6—figure supplement 1A*) the cells were lysed with 200 µg/ml lysozyme (Sigma cat. #L2879) in low-salt buffer (10 mM Tris–HCl pH 8.0, 50 mM NaCl, 0.5 mM *tris* (2-carboxyethyl) phosphine (TCEP), 1 mM MgCl$_2$, 0.1 mg/ml DNase, 0.2% (vol/vol) Triton X-100, protease inhibitors) for 15 min at 25°C. The lysates were cleared by centrifugation and incubated with Ni-NTA agarose beads (20 µl per 400 µl lysate) for 2 hr at 4°C. The beads were washed with high-salt buffer (10 mM Tris–HCl pH 8.0, 500 mM NaCl, 0.5 mM TCEP, 1 mM MgCl$_2$) and proteins were eluted in SDS sample buffer.

Large-scale purification of pBpa-derivatized BiP proteins for crosslinking experiments was performed by Ni-affinity chromatography as described above in the 'protein purification' section for N-terminally His6-tagged BiP proteins with the exception that after binding of the proteins to the Ni-NTA affinity matrix the beads were additionally incubated with 67 µg/ml RNase A (Sigma), 67 mU/ml RNase T1 (Roche) (to minimize UV crosslinks to bacterial RNA that co-purifies with BiP) and protease inhibitors in HKM buffer for 30 min at 25°C. The concentrated eluate after affinity chromatography was supplemented with 5 mM ATP and further purified on a Superdex 200 10/300 GL gel-filtration column equilibrated with HKM. Proteins eluting with the same retention time as monomeric BiP were aliquoted and stored at −80°C.

BiP-3xFLAG (2 µM) was pre-incubated with 0.5 mM ATP (0.05 mM in final reaction) for 5 min at 25°C in HKM buffer and diluted 10-fold (0.2 µM in final reaction) into buffer containing (untagged) BiP$^{V461F}$ into which pBpa had been incorporated site-specifically (2 µM final). The mixtures were further incubated for 16 hr at 25°C to allow for efficient formation of stable *trans* BiP complexes, following hydrolysis of the ATP. Where indicated samples were then exposed to UV light using a Stratalinker 2400 UV crosslinker (Stratagene) for 20 min on ice and analyzed by SDS-PAGE and immunoblotting with FLAG M2-specific antibodies.

## Limited proteolysis of BiP in presence of nucleotide

Limited proteolysis of BiP was performed as previously described (*Wei et al., 1995*). Briefly, 10 µg of BiP proteins were digested in HKM buffer with 2 µg Proteinase K (Fisher Scientific cat. # BP1700) in a final volume of 65 µl at 37°C with or without added nucleotides as indicated. The reactions were stopped by addition of 5 mM PMSF and incubated for 30 min on ice. The digested samples together with undigested controls were analyzed by SDS-PAGE and Coomassie-staining.

## Mammalian cell culture

CHO-K1 cells were cultured at 37°C and 5% $CO_2$ in Nutrient mixture F-12 Ham (Sigma) supplemented with 10% (vol/vol) serum (FetalClone II; HyClone), 1 × Penicillin-Streptomycin (Sigma) and 2 mM L-glutamine (Sigma). Cells were transfected with 5 µg of indicated plasmid DNA per 10 cm dish using Lipofectamine LTX (Life Technologies) according to manufacturer's instructions. The medium was exchanged after 24 hr and experiments were performed 36 hr after transfection. For ER calcium measurements (*Figures 9* and *10*) cells were electrotransfected with plasmid DNA (10 µg per 1 × 10⁶ cells) encoding the D1ER cameleon probe (see '*Supplementary file 1*') using the Neon transfection system (Life Technologies) according to the manufacturer's protocol. Fluorescence lifetime measurements were performed 24 hr after transfection as described in (*Avezov et al., 2013*) and graphs were generated with Prism 6 (GraphPad Software, Inc.).

AR42j cells were cultured at 37°C and 5% $CO_2$ in DMEM (Sigma) supplemented with 10% (vol/vol) serum (FetalClone II, HyClone), 1 × Penicillin-Streptomycin (Sigma), 2 mM L-glutamine (Sigma), 1 × non-essential amino acids (Sigma) and 50 µM β-mercaptoethanol (Gibco; Life Technologies). Differentiation into exocrine cells (*Logsdon et al., 1985*) was induced by treatment with 100 nM dexamethansone for 48 hr before the experiments.

All experiments with untransfected cells were performed at cell densities of 60–80% and the medium was exchanged 4 hr before the experiments. For pharmacological treatments the drugs were first diluted into pre-warmed culture medium, mixed and immediately applied to the cells by medium exchange. Unless indicated otherwise the following final concentrations were used: 100 µg/ml cycloheximide, 0.5 µM thapsigargin, 1 mM DTT, 2.5 µg/ml tunicamycin, 4 mM azetidine-2-carboxylic acid, 0.5 mM novobiocin, 4 µM cholecystokinin, 15 nM AP20187, 6 µM U73122, 10 µM A23187, 10 µg/ml brefeldin A.

## Mammalian cell lysates

Mammalian cells were grown on 10 cm dishes and treated as indicated. At the end of each experiment the dishes were immediately placed on ice and washed twice with ice-cold PBS (137 mM NaCl, 2.7 mM KCl, 10 mM $Na_2HPO_4$, 1.8 mM $KH_2PO_4$). The cells were then detached with a cell scraper in PBS containing 1 mM EDTA, transferred to a 1.5 ml reaction tube and sedimented for 5 min at 370 *g* at 4°C. The cells were lysed in HG lysis buffer (20 mM HEPES-KOH pH 7.4, 150 NaCl, 2 mM $MgCl_2$, 10 mM D-glucose, 10% [vol/vol] glycerol, 1% [vol/vol] Triton X-100) supplemented with protease inhibitiors (1 mM PMSF, 2 µg/ml pepstatin, 2 µg/ml leupeptin, 4 µg/ml aprotinin) for 10 min on ice and centrifuged at 21,000 *g* for 10 min at 4°C. The protein concentrations of the supernatants were determined by the Bradford method using BIO-RAD protein assay (BioRad) and normalized with lysis buffer. Afterwards the proteins were denatured by addition of SDS sample buffer and heating for 10 min at 75°C followed by separation on 12% SDS polyacrylamide-gels (30 µg lysate/lane).

For detection of endogenous BiP by native polyacrylamide gel electrophoresis (native-PAGE) the cells were lysed in HG lysis buffer containing 2 × protease inhibitors (2 mM PMSF, 4 µg/ml pepstatin, 4 µg/ml leupeptin, 8 µg/ml aprotinin) with or without 100 U/ml hexokinase (Type III from baker's yeast; Sigma) as described above and samples were loaded immediately on native or SDS-polyacrylamide

gels. Phosphatase inhibitors (10 mM tetrasodium pyrophosphate, 100 mM sodium fluoride, 17.5 mM β-glycerophosphate) were added to the lysis buffer in experiments where phosphorylated eIF2α was detected.

## Native polyacrylamide gel electrophoresis (native-PAGE)

A discontinuous Tris-glycine polyacrylamide gel system consisting of a 4.5% stacking gel and a 7.5% separation gel was used to separate purified protein or mammalian cell lysates under non-denaturing conditions. To detect BiP oligomers the gels were run in a Mini-PROTEAN electrophoresis chamber (BioRad) in running buffer (25 mM Tris, 192 mM glycine, pH ~8.8) at 120 V for 2 hr when cell lysates were applied or for 1:45 hr when His6-tagged purified BiP proteins were analyzed. The proteins were then stained with InstantBlue Coomassie solution (Expedeon) or transferred to a polyvinylidene difluoride (PVDF) membrane in blotting buffer (48 mM Tris, 39 mM glycine, pH ~9.2) containing 0.04 (wt/vol) SDS for 16 hr at 30 V for immunodetection. After the transfer the membrane was washed for 20 min in blotting buffer supplemented with 20% (vol/vol) methanol before blocking. 7 μg of purified protein was loaded per lane on a native gel to detect BiP oligomers by Coomassie staining and 30 μg mammalian cell lysates were loaded per lane to detect endogenous BiP by immunoblotting.

## Immunoblot analysis

Proteins were separated by SDS-PAGE or native-PAGE and transferred to PVDF membranes. The membranes were blocked with 5% (wt/vol) dried skimmed milk in TBS (25 mM Tris–HCl pH 7.5, 150 mM NaCl) and probed with the indicated primary antibodies and IRDye fluorescently labeled secondary antibodies (LiCor). The fluorescence signals were detected with the Odyssey near-infrared imager (LiCor). Where indicated densitometric quantification of immunoblot signals was performed with ImageJ (NIH). Primary antibodies and antisera against hamster BiP (chicken α-BiP; [*Avezov et al., 2013*], eIF2α (mouse α-eIF2α; [*Scorsone et al., 1987*]), phosphorylated eIF2α (rabbit α-eIF2α Phospho [pS51]; Epitomics cat. # 1090-1), the FLAG-tag [mouse ANTI-FLAG M2; Sigma cat. #F1804], human IgG [goat-α-human IgG; Sigma cat. #I1886], amylase (goat α-amylase [C-20]; Santa Cruz Bio-technology cat. # sc-12821) and the C-terminus of hamster BiP [rabbit against GEEDTSGKDEL peptide; [*Hendershot et al., 1995*]) were used.

## Co-immunoprecipitations and protein A pulldown

For co-immunoprecipitation of endogenous BiP with the FLAG-tagged null Hong Kong variant of α1-antitrypsin (AAT-NHK-3xFLAG) CHO-K1 cells were transfected with the method described above and treated without or with 1 μM thapsigargin for 5 min before lysis in HG lysis buffer supplemented with protease inhibitors and hexokinase. The lysates were cleared twice, normalized and equal volumes of the lysates were incubated with 25 μl ANTI-FLAG M2 beads (Sigma cat. # A2220) for 1 hr at 4°C, respectively. The beads were then recovered by centrifugation for 1 min at 8200 g and washed three times for 5 min at 4°C with HG lysis buffer without hexokinase. Bound proteins were eluted in 40 μl HG lysis buffer supplemented with 0.125 ng/μl 3xFLAG peptides (Sigma cat. #F4799) for 40 min at 4°C and the beads were removed by centrifugation. The proteins in the eluate supernatants were denatured by addition of SDS sample buffer and heating for 10 min at 75°C. Equal volumes of the samples were loaded on a 12% SDS polyacrylamide-gel and AAT-NHK-3xFLAG and endogenous BiP were detected by immunoblotting with FLAG- and BiP-specific antibodies. Samples of the normalized lysates (30 μg) were loaded as an 'input' control.

To analyze the interaction of endogenous BiP with immunoglobulin heavy chain (IgHC, [*Liu et al., 1987*], [*Hendershot et al., 1995*]) transfected CHO-K1 cells were treated and lysed as described above. Equal volumes of the normalized cleared lysates were then incubated with 20 μl of Protein A Sepharose 4B beads (Invitrogen cat. # 10–1042) for 1 hr at 4°C, the beads were sedimented by centrifugation for 2 min at 850 g and washed three times with HG lysis buffer. The proteins were eluted in 50 μl 2x SDS sample buffer for 10 min at 75°C and equal volumes of the samples were analyzed by SDS-PAGE and immunoblotting with antibodies directed against hamster BiP and human IgG.

The interaction of BiP with amylase was studied in AR42j cells. The cells were therefore exposed for 48 hr to 100 nM of the steroid hormone dexamethasone to enhance production of secretory proteins followed by treatment with or without 4 μM cholecystokinin for 5 min and lysis in

HG buffer supplemented with protease inhibitors and hexokinase. Equal volumes of the cleared and normalized lysates were incubated for 2 hr at 4°C with 20 μl UltraLink Hydrazine Resin (Pierce cat. # 53,149) on which BiP-specific chicken IgY antibodies have been covalently immobilized according to the manufacturer's instructions. The beads were recovered by centrifugation for 1 min at 1000 $g$ and washed three times with HG lysis buffer. The proteins were eluted in 50 μl 2 × SDS sample buffer and equal volumes of the samples were applied to SDS-PAGE. Endogenous BiP as well as amylase were detected by immunoblotting. Samples of the normalized lysates (30 μg) were loaded as an 'input' control. For the reverse experiment, the lysates were incubated with 15 μl Protein G Sepharose 4B beads (Invitrogen cat. # 10–1242) for 30 min at 4°C (pre-clearing) followed by incubation for 2 hr at 4°C with 15 μl beads coupled with amylase-specific or non-specific control antibodies. Afterwards the beads were sedimented by centrifugation for 2 min at 850 $g$, washed twice with HG lysis buffer supplemented with or without 0.5 mM ATP for 5 min at 4°C and twice with HG lysis buffer with or without 1 mM ATP. Proteins were then eluted in 2× SDS sample buffer, separated by SDS-PAGE and amylase and BiP were detected by immunoblotting.

## Acknowledgements

We thank Janet Deane (CIMR, Cambridge) for assistance with SEC-MALS, Matthias Mayer (ZMBH, Heidelberg) for the gift of the pCA528 plasmid and Peter Schultz (Scripps Institute) for the gift of the pSup-BpaRS-6TRN plasmid.

Supported by grants from the Wellcome Trust (Wellcome 084812/Z/08/Z) the European Commission (EU FP7 Beta-Bat No: 277713), a Wellcome Trust Strategic Award for core facilities to the Cambridge Institute for Medical Research (Wellcome 100140) and a US Public Health Service grant NIH-GM54068 (to LH). DR is a Wellcome Trust Principal Research Fellow.

## Additional information

### Competing interests

DR: Reviewing editor, *eLife*. The other authors declare that no competing interests exist.

### Funding

| Funder | Grant reference | Author |
| --- | --- | --- |
| European Commission (EC) | EU FP7 Beta-Bat No: 277713 | David Ron |
| Wellcome Trust | Wellcome 084812/Z/08 | David Ron |
| Wellcome Trust | Wellcome 100140 | David Ron |
| US Public Health Service | NIH-GM54068 | Heather P Harding |

The funder had no role in study design, data collection and interpretation, or the decision to submit the work for publication.

### Author contributions

SP, Led the project, designed and conducted and interpreted the bulk of the experiments and wrote the paper; JEC, Initiated the project and established the early crucial in vitro evidence of linker engagement as the basis for BiP oligomerization; AC-C, Purified proteins; EA, Performed measurements of ER calcium levels; EM, JP, Contributed to experiments on in vivo oligomerization of BiP and to editing the paper; LMH, Contributed critical insight into the design and interpretation of the experiments and to the writing of the paper; HPH, Implemented the techniques for site-specific modification of recombinant BiP in *E. coli* and carried out the site-specific crosslinking experiments; DR, Conceived and oversaw the study as a whole, wrote the paper and designed and constructed expression plasmids for the study

### Author ORCIDs

Steffen Preissler, http://orcid.org/0000-0001-7936-9836
Elena Miranda, http://orcid.org/0000-0002-0586-8795
David Ron, http://orcid.org/0000-0002-3014-5636

## Additional files

**Supplementary file**
• Supplementary file 1. Plasmids used.

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
