## [Decision Letter]

20 January 2015: Thank you for choosing to send your work entitled “Physiological modulation of BiP activity by *trans*-protomer engagement of the interdomain linker” for consideration at *eLife*. Your full submission has been evaluated by Vivek Malhotra (Senior Editor), Peter Walter (Guest Reviewing editor), and three peer reviewers, and the decision was reached after discussions between the reviewers. Based on our discussions and the individual reviews below, we regret to inform you that your work will not be considered further for publication in *eLife*. We would like to encourage you, however, to submit a substantially refocused manuscript as described below, which if you chose to do so would be subject to a new round of review.

As you will see below, while the level of enthusiasm of the reviewers varied, all agreed that the experiments are cleverly designed and executed, and that the underlying hypothesis is an attractive and important one. In the current manuscript the mechanistic studies in vitro (Figures 1, 2 and 3) carry the weight of the conclusions, and the in vivo work complements largely by being consistent with the hypothesis. It is the importance to the chaperone/protein folding field of the linker-mediated oligomerization that can lift this paper to *eLife* standards, especially if more direct evidence of that interaction could be included.

Reviewer #1:

Summary:

In this manuscript the authors analyze oligomerization of Bip in vitro and in vivo. Using bacterial protease SubA, specific for the interdomain linker, the authors explore the role of this domain in oligomerization of Bip. Adding SubA to CHOK1 lysates in the presence of ATP increased its susceptibility to interdomain cleavage and depolymerization. Gel exclusion chromatography using purified WT Bip and mutants of SBD and linker domains further established the role for the substrate binding domain (SBD) and interdomain in oligomerization. This confirmed previous observations on ATP mediated Bip depolymerization and suggest a role for the interdomain in this process. The authors then go on to other experiments to analyze the arrangement of protomers in the Bip oligomer. Through analysis of combinations of SBD mutant (V461F), interdomain mutant (ADDA) and double mutant (ADDA-V461F) either purified or transfected into cells followed by gel filtration and native gel electrophoresis. The authors suggest *trans*-protomer engagement of the interdomain with the SBD as the mechanism of Bip oligomerization in vitro and in vivo that is independent of the helical lid. Induction of protein misfolding by most methods (dithiothreitol (DTT), cycloheximide (CHX), activation of Fv2E-PERK, tunicamycin {trade mark, serif} etc) induced depolymerization of Bip possibly to make it available to bind to unfolded proteins. However, it was observed that ER calcium depletion promoted appearance of oligomers by promoting dissociation from its substrates but the cellular function/significance for this shift in equilibrium is unknown.

Overall:

The major point of this paper is a model proposed for Bip oligomerization by a trans-protomer engagement of one SBD with the interdomain of the adjacent protomer. It is still not clear that one can convincingly conclude this model based on the results presented here. The data are interpreted as being consistent with previous findings/models but alternative explanations are possible and not considered. For example, in the subsection “In vitro BiP oligomerization by cross-protomer engagement of the interdomain linker at *trans* substrate binding sites” it is stated that “Importantly, addition of ATP has no effect on migration of the ADDA mutant, consistent with abolition of allosteric coupling between the domains...”. It is difficult to make these conclusions from studying a single mutant protein. One experiment to support the *trans*-protomer engagement rather than a burying effect of the interdomain in the presence of ATP would be to directly show interaction of the interdomain with the SBD. The findings are consistent with crystal structures of a DnaK family member and biophysical analyses showing the interdomain linker binds the SBD of an adjacent protomer (12; 2). These latter studies are much more thorough and definitive than the present analysis.

Overall, the first three figures are fairly convincing that analyze purified Bip in vitro, if the authors can further successfully address the following alternative explanations:

In Figure 1, there is possibility that the ATP/ADP ratio or ADP concentration affects oligomer formation since the ADP-bound form was shown to be a more stable form of Bip.

The reviewer wonders if the ATP or ATP/ADP ratio affects SubA protease activity? This can be easily checked out using the linker peptide substrates as cleavage readout. Figure 4 seems to support this scenario that ATP addition increases Bip cleavage (compare lane 2 and 4).

Figure 2, controls are needed for AMP and ADP, as well as another nucleotide (such as GTP) as negative controls.

In contrast and unfortunately, the cellular studies (Figures 4, 5, 6, 7, 8, 9 and 10) have many limitations that preclude any significant interpretation. The findings are primarily descriptive and do not significantly extend our mechanistic understanding of Bip oligomerization.

Major comments:

1) Most of the observations reported in this paper are consistent with many previous results on the HSP70 family members, generally, and on Bip oligomerization, specifically. The amount of new insight is fairly limited and in some cases the data are consistent with the model, but the data do not significantly support the model. For example, the oligomerization of Bip and effect of ATP and peptide binding have previously been described years ago (19; 13). The latter paper actually used a similar technique, enterokinase sensitivity to cleave the NBD from the SBD and measured the impact of ATP binding and peptide binding on oligomerization which produced very similar conclusions drawn from the present study, although the latter paper was not referenced.

2) The results primarily rely on sensitivity of wildtype Bip and mutant Bip molecules to the protease SubA. It is unclear what cleavage susceptibility by SubA means. Allosteric changes may preclude cleavage by SubA without changing exposure of the interdomain linker.

3) Figure 4—figure supplement 1, it is clear that at high concentration, Bip has intrinsic oligomerization propensity regardless of ATP concentration. How would this property affect Bip oligomerization during sudden change in ER ionic strength, such as thapsigargin-mediated Ca2+ depletion?

4) The results of the findings that show calcium depletion promotes oligomer formation are inconsistent with current notions that calcium is required for chaperone function and protein folding. Thus, calcium depletion should cause more misfolding and deplete oligomers. This discrepancy needs to be clarified to understand the impact of protein misfolding vs. calcium on oligomer formation as proposed in Figure 10. To support the model depicted in Figure 10, the authors need to show a native gel of Bip oligomer formation as a function of escalating Ca2+ concentrations in vitro.

5) There are inconsistencies with the data from different experiments that are difficult to explain. Some of these are listed below.

A) In Figure 1, lanes 1-3 minus ATP; there are a number of intermediates of Bip that may represent Bip binding to endogenous peptide ligands. These are not present with ATP and disappear with SubA cleavage. From all of the experiments it is not clear what fraction of the Bip analyzed is oligomeric vs. bound to peptide ligands of different sizes that would migrate as a heterogeneous background.

B) From analysis of the gel in Figure 1, in the presence of ATP there seems to be less cleaved products especially at lower concentrations of SubA (0.03, 0,08..) compared to minus ATP. Since this is the basis of the whole assay, please clarify.

C) It is difficult to know how different mutations in Bip impact overall structure (ADDA; V461F).

D) Figure 8 is missing an important control of ATP alone.

E) Figure 6 shows a 'B' form that is assumed to be ADP-ribosylated/AMPylated, although this is not directly demonstrated. This is inferred from use of novobiocin to inhibit ADP-ribosylation. Why does Figure 6 lane 3 show the 'B' form in the presence of novobiocin and cycloheximide?

F) The authors conclude that stresses that inhibit protein synthesis eliminate the oligomers and promote formation of the 'B' form. Then the authors study different ER stresses, Tm, DTT, Aze to show that they also cause depletion of the oligomers, but the 'B' form does not appear. These stresses should also induce eIF2a phosphorylation and inhibit protein synthesis. However, analyses of eIF2a-P and protein synthesis were not directly studied. If the oligomers disappear, where do they go? These studies should be performed with and without ATP incubation in vitro prior to gel analysis to dissociate potential Bip ligands.

Reviewer #2:

The anticipated wide fluctuation in client load within the endoplasmic reticulum (ER) necessitates systems to maintain folding homeostasis. Given the limitations and costs of transcriptional systems like the UPR to maintain homeostasis, a new focus has been placed on identification of more rapid post-translational adaptations that likely exist to adapt to changes in the ER protein load. This manuscript by Preissler et al. furthers our understanding of post-translational adaptation to changes in ER homoestasis and focuses on the molecular basis of oligomerization of the molecular chaperone BiP. Several prior studies have identified the propensity of Hsp70 s to oligomerize, however this study is one of the few to focus on a molecular basis for oligomerization. The manuscript convincingly presents data supporting a mechanism for *trans-*engagement of BiP protomers to facilitate oligomerization. Using various drug treatments in cells, the authors propose a model for a reciprocal relationship between the burden of unfolded proteins and BiP oligomers, and a slower equilibration between oligomers and inactive, covalently modified BiP.

Compelling data for *trans*-engagement of adjacent protomers through the substrate binding site of one protomer and the linker of the second protomer are presented using a variety of in vitro and in vivo methods. Such a mechanism is consistent with a prior structure of Hsp70 protomers engaged in a similar *trans*-engagement. in vivo data suggest physiological trigger(s) exist for oligomerization, which may serve to create a pool of inactive yet readily mobilizable BiP.

A few minor comments:

Is the ADDA mutant previously described? In the subsection “In vitro BiP oligomerization by cross-protomer engagement of the interdomain linker at trans substrate binding sites”, the authors describe the mutant as “abolishing the linker's role in allostery, disrupting the SubA cleavage site and linker's ability to serve as a BiP substrate, all whilst preserving the intrinsic ability of BiP's SBD to bind other substrates.” There is no reference given, suggesting that the data supporting these statements are derived from the later experiments of this manuscript. If this is the case, it would help to clarify that this mutant was made with the expectation of these effects that were then confirmed by the authors?

The main focus of the manuscript is the molecular basis for *trans* oligomerization. However, the authors do attempt to create a larger model depicting the interplay between calcium, covalent modification, monomer, and oligomers. The model as proposed is consistent with the data presented, however is seems a few simple experiments could strengthen the arguments for the transitioning between these species. First, does calcium addition to pure protein prevent oligomerization (or is this perhaps as argued related to a shift in client load in the ER?)? Along the same lines, does calcium depletion impact covalent modification of BiP? Second, do mutants in R470or R492, that prevent ADP-ribosylation, impact oligomerization?

Reviewer #3:

This is a beautifully written and carefully conceived study of the oligomerization of BiP upon several physiological perturbations that alter the folding status of the ER. The authors' original observation – that the interdomain linker became sequestered from proteolysis concurrent with a shift to a higher molecular weight complex – led them to a model for BiP oligomerization that is based on binding of the linker of one monomer to the substrate-binding cleft of another monomer, to form a concatenated oligomer. The authors carry out a large number of clever experiments to test this model, and all the data they present are consistent with the proposed mode of interaction. The case is therefore persuasive. The other work presented in this paper probes the impact of the folding state of the ER and the calcium content on oligomerization, and all of this work is well-conceived and implemented. The expertise of the senior authors is clearly manifest in the way the experiments are designed and interpreted. The picture that clearly emerges is that oligomerization is part of an interwoven network of regulatory mechanisms to enable the ER to respond to changing folding demands.

Despite the compelling significance and overall clarity of the work presented, none of the experiments in the paper, even taken with the previous crystal structures of oligomers, provides direct evidence for the interaction between the linker of one molecule and the substrate-binding domain of another, and it is a small but non-zero leap-of-faith to accept that this interaction forms the basis for oligomerization. This is a weakness of this otherwise top-notch paper: And it would seem the authors are equipped to address this question by cross-linking and mass spectrometry experiments. Another improvement of the paper that seems quite do-able is more direct characterization of the state of oligomerization (dimer, trimer, etc). It should be possible to do light glutaraldehyde crosslinking and SDS gel characterization of resulting products to determine what species are present in the SEC peaks and native gel bands.

Minor comments:

1) I would recommend taking the very helpful schematic in Figure 5—figure supplement 1 and putting into the text early in the Results section, as part of a little new section “Design of experiments”. Without this, the number of constructs and mutant BiP versions and their role in the experiments becomes a bit daunting for the reader. It wouldn't hurt to show the products expected from each experiment in the figure, and the molecular weights anticipated. The reader would then be prepared for the results in the primary data.

2) The relationship of the oligomerization story to the ADP ribosylation and AMPylation modifications is a bit murky. In the Discussion, the speculation is interesting. But earlier these modifications are presented and only seem to confuse the story presented.

3) Results: the wording of the subheading about “Physiological release” of calcium is confusing. Is the Ca level in the ER being depleted?

4) Figure 4 legend: a word is missing between "that" and "SubA".

5) Experiments with transfection of BiP variants in CHO cells: how substantial is the over-expression of the transfected BiP? And how much will this affect the oligomerization/folding balance? I don't believe these issues were discussed.

A few additional points came up during the discussion among the reviewers and may be worthy of consideration:

1) the ADDA mutant seem to form oligomers by size exclusion chromatography in Figure 2 and native gel electrophoresis in Figure 5—figure supplement 1. This seems inconsistent with the lack of endogenous BiP interaction with the ADDA mutant shown in lane 5 and 6 in Figure 5. This should be explained.

2) In Figure 6 lane 2, oligomeric states of BiP disappear when modified by ADP ribosylation. The authors do not explain the discrepancy with Freiden et al. who showed that oligomeric forms of BiP harbor the modification. Perhaps more importantly, Hu et al. (PMID: 19808260) indicate that cycloheximide treatment renders BiP insensitive to SubAB cleavage in intact B cells. Yet, work in this manuscript suggests the opposite. These points should be addressed.

3) The authors indicate that Ca++ depletion was shown earlier to interfere with BiP's ability to engage in substrate binding. If the basis of BiP's oligomerization is through its binding to its linker as a substrate, how can the oligomeric state persist upon Ca++ depletion? The attempt to explain these findings in the manuscript is not clear.

[Editors’ note: further revisions were requested before acceptance, as outlined below.]

14 July 2015: Thank you for submitting your work entitled “Physiological modulation of BiP activity by *trans*-protomer engagement of the interdomain linker” for peer review at *eLife*. Your submission has been favorably evaluated by Vivek Malhotra (Senior editor), Peter Walter (Guest Reviewing editor), and three reviewers.

The reviewers have discussed the reviews with one another and the Reviewing editor has drafted this decision to help you prepare a revised submission.

Summary:

This work on BiP oligomerization is the most definitive and well-articulated description of the “binding your own interdomain” model for an Hsp70. While the physiological role of this process remains uncertain, the models proposed provide hypotheses for further work. The quality of the science and the writing are top-notch.

In this revised manuscript, the authors address previous concerns. Prior reviews focused on two primary limitations: a lack of direct evidence for *trans*-engagement of the BiP linker and an absent discussion of the potential for allosteric effects of the mutants (specifically ADDA) used within the study. The authors now include several experiments that aim to decouple the impact of allosteric mutations from linker disruption. The new data are consistent with the original model and provide new mutants that allow one to ascribe certain features to the linker, and not decoupling. The authors incorporate new experiments making use of the incorporation of non-natural amino acid cross-linkers, with the goal of providing more direct evidence for *trans*-engagement of the linker domain. The authors conclude that only cross-linkers incorporated within the linker can engage (cross-link with) a *trans* wild-type BiP. At face value, these data provide a more direct link for *trans*-linker engagement.

Essential revisions:

The following points need to be considered.

1) The authors include new data with mutants T37G and G226D and photocrosslinking experiments to prove the role of the interdomain in oligomerization. This still does not help to explain any possible allosteric effects of ADDA mutant itself. Is there a reason why ADDA was not included in Figure 5 to be compared with WT and other mutants side by side?

2) Since calcium has no effect in vitro, is it possible that what the authors are considering Bip oligomers in vivo are in fact Bip in complex with other endogenous substrates? This possibility needs to be considered.

3) The authors failed to examine effect of other nucleotides (AMP/ADP/ATPγS/GTP/GDP, etc), invalidating the claim that Bip aggregate formation is dependent on ATP concentration. Quote here, “As the reviewer well knows, ATP and ADP are expedients to shift the monomer oligomer equilibrium.” It is unclear what this statement means regarding the requirement for nucleotide and ATP hydrolysis in the oligomerization. Is the ATPase activity important for oligomer-monomer conversion?

---

## [Author Response]

24 May 2015: An earlier version of this manuscript was reviewed and though that earlier version was rejected, the reviewers and editors recognized the importance and interest of the problem under study and encouraged us to try and improve the manuscript and, provided we were able address the important limitations of the earlier version, submit an improved manuscript for review.

The most important limitation of the earlier version was deemed to be the strength of the evidence for the specific architecture of BiP oligomers, which we propose to consist of adjacent BiP protomers engaging each other’s hydrophobic interdomain linker as a typical substrate in their substrate binding domain. Whilst all three reviewers recognized that the evidence provided for such an arrangement of protomers was consistent with the model proposed, it was pointed out that we had failed to adequately rule out allosteric effects of the mutants applied to the study of the problem or to provide direct evidence for engagement of the interdomain linker.

The revised version of the manuscript contains two entirely new figures that address these issues:

It was pointed out that the previous version of the manuscript failed to take into account the possibility that the mutant BiPADDA, bearing an altered interdomain linker, is impaired in oligomerization not because the mutations lowered the affinity of the interdomain linker for the substrate binding domain of adjacent protomers, but because disruption of the interdomain linker locked BiP into a configuration that allosterically impaired oligomerization. We recognize the validity of this criticism, as it is known that the interdomain linker is required for Hsp70 chaperones to assume the domain-coupled configuration. New Figure 5 and its accompanying two supplements confirm that BiPADDA is locked in an uncoupled configuration, as expected. However, other mutations that favor the uncoupled configuration (BiPT37 G and BiPG226D) exhibit enhanced oligomerization, as suggested by our model, whereas BiPADDA exhibits impaired oligomerization. These observations uncouple the allosteric effect of the BiPADDA mutation from its effect on oligomerization.

To directly examine the engagement of the interdomain linker as a substrate in oligomer formation, we followed the suggestions of reviewer #3 and exploited the system developed by the Schultz lab for genetically incorporating an unnatural crosslinkable amino acid site-specifically into BiP. New Figure 6 and its supplements shows conspicuous crosslink formation when the zero-length crosslinkable para- benzoyl-L-phenylalanine amino acid was incorporated in place of residues in BiP’s interdomain linker, but not when incorporated at a distance from the interdomain linker. These findings directly implicate the interdomain linker in BiP oligomerization.

In addition to these key improvements, we have taken onboard other points of critique. These have been addressed by providing additional data (e.g. new Figure 2—figure supplement 1) by rearranging existing data (e.g. new Figures 1 and 12) or by editorial changes. A point-by-point list of the critiques and changes made follows.

Reviewer #1:

*[…] In*
Figure 1*, there is possibility that the ATP/ADP ratio or ADP concentration affects oligomer formation since the ADP-bound form was shown to be a more stable form of Bip.*

*The reviewer wonders if the ATP or ATP/ADP ratio affects SubA protease activity? This can be easily checked out using the linker peptide substrates as cleavage readout.*
Figure 4
*seems to support this scenario that ATP addition increases Bip cleavage (compare lane 2 and 4).*

We believe that the figure as presented makes the point that in samples in which the ATP was converted to ADP a substantial fraction of BiP is found in a configuration whereby its interdomain linker is not digested by SubA.

The reviewer’s suggestion that the effects of nucleotide on the substrate be uncoupled from their effects on the protease, by measuring the cleavage of a short peptide corresponding in sequence to the interdomain linker of BiP is conceptually sound. However we are not aware that a short peptide corresponding in sequence to the interdomain linker of BiP would serve as a substrate for SubA and even if it did, and if nucleotide manipulation failed to affect cleavage, this would constitute a weak control experiment as doubt would remain as to whether or not a short peptide could capture a regulatory effect of nucleotide. Furthermore, subtilisin and the related chymotrypsin class of proteases are not known to bind nucleotides or to be influenced by their presence. Thus, whilst it is formally possible that nucleotide concentrations affect some aspect of SubA activity, we interpret the observations presented in Figure 4, whereby ATP depletion promotes BiP^V461F^ digestion whilst attenuating the digestion of BiP^WT^, to argue that the effects of nucleotide are, at the very least substrate-specific and more in keeping with action on the substrate than on the protease.

Figure 2*, controls are needed for AMP and ADP, as well as another nucleotide (such as GTP) as negative controls.*

As the reviewer well knows, ATP and ADP are expedients to shift the monomer oligomer equilibrium. It is unclear to us what specific shortcoming in the interpretability of the data presented in Figure 2 would be overcome by the use of other nucleotides as suggested by the reviewer.

Major comments:

*1) Most of the observations reported in this paper are consistent with many previous results on the HSP70 family members, generally, and on Bip oligomerization, specifically. The amount of new insight is fairly limited and in some cases the data are consistent with the model, but the data do not significantly support the model. For example, the oligomerization of Bip and effect of ATP and peptide binding have previously been described years ago (*[19]*;*
[13]*). The latter paper actually used a similar technique, enterokinase sensitivity to cleave the NBD from the SBD and measured the impact of ATP binding and peptide binding on oligomerization which produced very similar conclusions drawn from the present study, although the latter paper was not referenced.*

We readily concede our observations to be consistent with those previously made by others. However, we believe that the critical point of our manuscript: That BiP oligomerization proceeds by trans-linker engagement and the exposition of the consequence of this architecture of the oligomers to their functional state, is a worthy novelty.

We thank the reviewer for drawing our attention to the Chevalier paper, which is now cited prominently. We believe the parallels between the use of enterokinase by Chavalier and colleagues and our application of SubA to be non-redundant. In that Chavalier engineered an enterokinase site into an exposed loop at the C-term of BiP’s NBD to separate the NBD and SBD, not to probe the disposition of the interdomain linker.

2) The results primarily rely on sensitivity of wildtype Bip and mutant Bip molecules to the protease SubA. It is unclear what cleavage susceptibility by SubA means. Allosteric changes may preclude cleavage by SubA without changing exposure of the interdomain linker.

We recognize the validity of this point and thank the reviewer for this critique. In response, we have examined in detail other mutations that mimic the allosteric effects of the BiP^ADDA^ (to lock BiP in the domain uncoupled configuration) and found that unlike the BiP^ADDA^ mutation, which interferes with oligomerisation, these domain-uncoupling mutations (BiP^T37G^ and BiP^G226D^) are more prone to oligomerization (new Figure 5 and its supplements).

*3)*
Figure 4—figure supplement 1*, it is clear that at high concentration, Bip has intrinsic oligomerization propensity regardless of ATP concentration. How would this property affect Bip oligomerization during sudden change in ER ionic strength, such as thapsigargin-mediated Ca2+ depletion?*

*4) The results of the findings that show calcium depletion promotes oligomer formation are inconsistent with current notions that calcium is required for chaperone function and protein folding. Thus, calcium depletion should cause more misfolding and deplete oligomers. This discrepancy needs to be clarified to understand the impact of protein misfolding vs. calcium on oligomer formation as proposed in*
Figure 10*. To support the model depicted in*
Figure 10*, the authors need to show a native gel of Bip oligomer formation as a function of escalating Ca2+ concentrations* in vitro*.*

BiP does indeed oligomerize in a concentration-dependent manner. The process is tuned by the presence of nucleotide with the ATP-bound state disfavoring oligomerization.

In vitro, there is no measureable effect of calcium on BiP oligomerization, in either the ATP state or ADP state or in the presence of the J-protein ERdj3 or the exchange factor GRP170 (not shown). In vivo, ER calcium depletion promotes such oligomerization. There are many possible ways in which this could come about. We humor ourselves with the idea that it could, for example, be promoted by a yet-to-be discovered ER-localized J-factor whose affinity for BiP as a substrate is increased selectively as [Ca^+2^]_ER_ declines. Given that the affinity of Hsp70 proteins for their substrates is governed by J-protein driven non-equilibrium mechanisms (15), such a scenario could account for BiP favoring itself as a client (oligomerizing) over other clients, as [Ca^+2^]_ER_ declines. For now, however, we must set aside these pleasing and plausible (to us) speculations and accept that this is an unresolved mystery. In this study [Ca^+2^]_ER_ depletion-mediated BiP oligomerisation is exploited merely to examine the consequences of in vivo modulation of the oligomeric status of BiP. We thank the reviewer for highlighting the need to address these considerations, which we have done in paragraph 8 of the revised Discussion.

The reviewer states that: “calcium depletion should cause more misfolding and deplete oligomers”. This assumes that misfolding is a proximal consequence of calcium depletion; could it be that formation of BiP oligomers under conditions of calcium depletion contributes to the unfolded protein load in the calcium-depleted ER by competing for client protein binding? This idea is consistent with early observations made by the Klausner lab (and reproduced here, Figure 11) in which ER calcium depletion promoted dissociation of complexes between BiP and a sentinel client (an unassembled subunit of the T-cell receptor, in that case, [59]). We readily concede that we have not proven that calcium depletion mediated oligomerization is the basis for the enhanced presence of misfolded proteins and the activation of the UPR in cells treated with thapsigargin, A23187 etc. And we readily accept that there may be circumstances (specific cell types with specific complement of clients and chaperones) in which calcium depletion has other deleterious effects on the protein folding environment in the ER. But we humbly suggest that our observations favor and alternative mechanism and are reason to re-examine long held views, such as those that impel the reviewer to favor a particular chain of causality.

5) There are inconsistencies with the data from different experiments that are difficult to explain. Some of these are listed below.

*A) In*
Figure 1*, lanes 1-3 minus ATP; there are a number of intermediates of Bip that may represent Bip binding to endogenous peptide ligands. These are not present with ATP and disappear with SubA cleavage. From all of the experiments it is not clear what fraction of the Bip analyzed is oligomeric vs. bound to peptide ligands of different sizes that would migrate as a heterogeneous background.*

*B) From analysis of the gel in*
Figure 1*, in the presence of ATP there seems to be less cleaved products especially at lower concentrations of SubA (0.03, 0,08..) compared to minus ATP. Since this is the basis of the whole assay, please clarify.*

We thank the reviewer for drawing our attention to this point and for giving us an opportunity to consider it explicitly. Indeed the impact of ATP is less on the kinetics of the cleavage observed early in the assay and more on its completeness. As explained in detail in the tenth paragraph of subsection “In vitro BiP oligomerization by cross-protomer engagement of the interdomain linker at *trans* substrate binding sites” of the revised manuscript, these observations are consistent with ATP binding having two opposing effects on interdomain linker exposure: (i) Protection against SubA by burial at the interface of the coupled NBD and SBD; a state that equilibrates relatively rapidly with the uncoupled linker-exposed state, leading ultimately to complete digestion by SubA (ii) Enhanced exposure to SubA by release of the linker from the SBD of adjacent protomers in the oligomer. In the absence of ATP the oligomers are very stable with a low off rate. Thus the addition of SubA to the ATP^minus^ sample leads to rapid cleavage of BiP monomers (and of the exposed protomer C from the oligomer, Figure 1) with the persistence of a fraction of SubA resistant oligomeric BiP, which equilibrates very slowly with monomers.

C) It is difficult to know how different mutations in Bip impact overall structure (ADDA; V461F).

These mutations are well characterized in the context of the highly related *E. coli* DnaK, furthermore, the intact ATP-mediated coupling of the NBD and SBD of BiP^V461F^ (Figure 4, right panel) argues that it is not an extensively broken machine. Similar considerations apply to BiP^ADDA^, which retains its ability to bind BiP^WT^ molecules (Figure 3) and whose NBD forms a protease resistant core, characteristic of the intact protein (New Figure 5—figure supplement 1).

*D)*
Figure 8
*is missing an important control of ATP alone.*

Cycloheximide was introduced into the basal state of this experiment (Figure 10 in the revised manuscript) to lower the background of BiP oligomers and favor detection of the effect of engagement of adenosine on oligomerization. It is unclear to us what specific shortcoming in the interpretability of the data presented in the figure would be overcome by the additional conditions suggested by the reviewer.

*E)*
Figure 6
*shows a 'B' form that is assumed to be ADP-ribosylated/AMPylated, although this is not directly demonstrated. This is inferred from use of novobiocin to inhibit ADP-ribosylation. Why does*
Figure 6
*lane 3 show the 'B' form in the presence of novobiocin and cycloheximide?*

We interpret the data (now in Figure 8) as showing rather conspicuously less “B” form in the sample from cells exposed to novobiocin (compare lanes 2 and lanes 3). Thus we believe that a strong case can be made that the “B” form present on the native gels here corresponds to the modified forms observed previously.

It is worth noting however that in vivo modified BiP has only been observed indirectly: As a species labeled with tritiated adenosine or orthophosphate (9), as an acidic species on IEF that is altered in its susceptibility to cleavage by SubA in vitro (10) and one that may account for the observation of SubA resistance to cleavage of BiP in cells starved for many (6) hours of methionine and cysteine (as in Figure 2 of (Hu et al., 2009), see comments of reviewer #3). Recently, there has been well-justified excitement by the discovery, for the first time, of an enzyme that modifies BiP (albeit by AMPylation, not ADP-ribosylation), but this modification too has been largely observed in vitro. Thus, our understanding of BiP modification is incomplete and very much a work in progress. Here we have tried to reconcile our finding to the prevailing concepts regarding the function of modified BiP, but it would be unwise to over-fit our data to these evolving concepts and we urge the reviewers too to maintain a modicum of flexibility in this regard.

*F) The authors conclude that stresses that inhibit protein synthesis eliminate the oligomers and promote formation of the 'B' form. Then the authors study different ER stresses, Tm, DTT, Aze to show that they also cause depletion of the oligomers, but the 'B' form does not appear. These stresses should also induce eIF2a phosphorylation and inhibit protein synthesis. However, analyses of eIF2a-P and protein synthesis were not directly studied. If the oligomers disappear, where do they go? These studies should be performed with and without ATP incubation* in vitro *prior to gel analysis to dissociate potential Bip ligands.*

A competition between oligomerization and substrate binding is consistent with fewer oligomers in cells in which protein folding in the ER is compromised directly.

Like other manipulations that inhibit protein synthesis, high levels of eIF2a phosphorylation (induced by pharmacological activation of a PERK derivative) will also promote the “B” form (formerly Figure 6, now Figure 8, lane 5 & 6), however, though endogenous PERK is activated in cells exposed to Tm, DTT, Aze, similar levels of eIF2a phosphorylation and translational attenuation are not achieved.

Reviewer #2:

Is the ADDA mutant previously described? In the subsection “In vitro BiP oligomerization by cross-protomer engagement of the interdomain linker at trans substrate binding sites”, the authors describe the mutant as “abolishing the linker's role in allostery, disrupting the SubA cleavage site and linker's ability to serve as a BiP substrate, all whilst preserving the intrinsic ability of BiP's SBD to bind other substrates.” There is no reference given, suggesting that the data supporting these statements are derived from the later experiments of this manuscript. If this is the case, it would help to clarify that this mutant was made with the expectation of these effects that were then confirmed by the authors?

We regret this omission in the earlier version of the manuscript. Similar mutations of the interdomain linker have been made and characterized years ago by (35) and their impact on SubA cleavage was assessed by (48). These references have now been included in the fourth paragraph of the subsection “In vitro BiP oligomerization by cross-protomer engagement of the interdomain linker at *trans* substrate binding sites”.

The main focus of the manuscript is the molecular basis for trans oligomerization. However, the authors do attempt to create a larger model depicting the interplay between calcium, covalent modification, monomer, and oligomers. The model as proposed is consistent with the data presented, however is seems a few simple experiments could strengthen the arguments for the transitioning between these species. First, does calcium addition to pure protein prevent oligomerization (or is this perhaps as argued related to a shift in client load in the ER?)? Along the same lines, does calcium depletion impact covalent modification of BiP? Second, do mutants in R470or R492, that prevent ADP-ribosylation, impact oligomerization?

The reviewer’s suggestions are most apt.

We have extensively studied the effects of calcium on BiP oligomerization in vitro in the presence or absence of nucleotide, purified J-protein and Grp170 (a Nucleotide Exchange Factor for BiP) (please see response to point 4 of reviewer 1), but no effect was observed. Thus, it would seem some component is missing from our in vitro assay. The implications of this are now discussed in paragraph 8 of the revised Discussion.

We recognize the interest of examining mutations that effect BiP modification on its ability to oligomerize. But as the R470K and R492K have other effects on substrate binding, we feel the experiment would not be powerful enough to deliver a clear answer to this interesting question and look forward to addressing this question with mutations in the enzyme(s) that modify BiP, when they become available.

Reviewer #3:

Despite the compelling significance and overall clarity of the work presented, none of the experiments in the paper, even taken with the previous crystal structures of oligomers, provides direct evidence for the interaction between the linker of one molecule and the substrate-binding domain of another, and it is a small but non-zero leap-of-faith to accept that this interaction forms the basis for oligomerization. This is a weakness of this otherwise top-notch paper: And it would seem the authors are equipped to address this question by cross-linking and mass spectrometry experiments. Another improvement of the paper that seems quite do-able is more direct characterization of the state of oligomerization (dimer, trimer, etc). It should be possible to do light glutaraldehyde crosslinking and SDS gel characterization of resulting products to determine what species are present in the SEC peaks and native gel bands.

We thank the reviewer for this important critique.

In response we have directly examined the engagement of the interdomain linker as a substrate in oligomer formation, exploiting the system developed by the Schultz lab for genetically incorporating an unnatural crosslinkable amino acid site specifically into BiP. New Figure 6 and its accompanying supplements show conspicuous crosslink formation when the zero- length crosslinkable para benzoyl _L_-phenylalanine amino acid was incorporated in place of residues in BiP’s interdomain linker, but not when incorporated at a distance from the interdomain linker.

We have also employed light gluteraldehyde crosslinking to probe the oligomeric status of BiP and the kinetics of the interconversion of the different species. These observations are presented in Figure 2—figure supplement 1.

Minor comments:

*1) I would recommend taking the very helpful schematic in*
Figure 5—figure supplement 1
*and putting into the text early in the Results section, as part of a little new section “Design of experiments”. Without this, the number of constructs and mutant BiP versions and their role in the experiments becomes a bit daunting for the reader. It wouldn't hurt to show the products expected from each experiment in the figure, and the molecular weights anticipated. The reader would then be prepared for the results in the primary data.*

We thank the reviewer for this comment. In response we have introduced the proposed schematic of BiP oligomerization as new Figure 1. We expect this will make the experimental design more accessible.

2) The relationship of the oligomerization story to the ADP ribosylation and AMPylation modifications is a bit murky. In the Discussion, the speculation is interesting. But earlier these modifications are presented and only seem to confuse the story presented.

We recognize that there is much that remains to be learned about BiP modification. But we think that a strong case can be made that the “B” form present on the native gels here corresponds to the modified forms observed previously and that our interpretation of the data in that light is unlikely to be grossly misleading. Beyond that, there is little we can be certain of (please see comments to point 5E of reviewer 2) and thus we have tried to reconcile our finding to the prevailing concepts regarding the function of modified BiP without over fitting our data to these evolving concepts.

3) Results: the wording of the subheading about “Physiological release” of calcium is confusing. Is the Ca level in the ER being depleted?

ER calcium depletion was documented by monitoring the change in fluorescent lifetime of an ER-localized calcium probe (Figures 9 and 10 in the revised version)

*4)*
Figure 4
*legend: a word is missing between “that” and “SubA”.*

We thank the reviewer for this comment. The omission has been corrected

5) Experiments with transfection of BiP variants in CHO cells: how substantial is the over-expression of the transfected BiP? And how much will this affect the oligomerization/folding balance? I don't believe these issues were discussed.

BiP overexpression is profoundly corrupting of cell physiology. For this reason, we have limited any conclusions drawn from the experiments to their ability to inform us on the oligomer’s architecture (which we believe remain valid in the face of corrupting over-expression), but have refrained from assigning any physiological significance to these experiments.

A few additional points came up during the discussion among the reviewers and may be worthy of consideration:

*1) the ADDA mutant seem to form oligomers by size exclusion chromatography in*
Figure 2
*and native gel electrophoresis in*
Figure 5—figure supplement 1*. This seems inconsistent with the lack of endogenous BiP interaction with the ADDA mutant shown in lane 5 and 6 in*
Figure 5*. This should be explained.*

The conspicuous species in BiP^ADDA^ co-migrates with the monomer, both on SEC and on native gel. But the defect in oligomerization, though conspicuous, is incomplete. Lower affinity interactions that involve other elements of BiP (that, partially dissociate during SEC or native gel electrophoresis) may account for these observations. The text in the second to last paragraph of the subsection “In vitro BiP oligomerization by cross-protomer engagement of the interdomain linker at *trans* substrate binding sites” has been revised to incorporate this point.

*2) In*
Figure 6
*lane 2, oligomeric states of BiP disappear when modified by ADP ribosylation. The authors do not explain the discrepancy with Freiden et al. who showed that oligomeric forms of BiP harbor the modification. Perhaps more importantly, Hu et al. (PMID: 19808260) indicate that cycloheximide treatment renders BiP insensitive to SubAB cleavage in intact B cells. Yet, work in this manuscript suggests the opposite. These points should be addressed.*

The observations of Hu et al. are consistent with our own showing the modified form of BiP is resistant to SubA cleavage (see Figure 6 of (10)). Indeed the pulse-chase labeling experiment that led Hu and colleagues to conclude that the newly synthesized BiP is selectively susceptible to SubA cleavage (Figure 2 in their paper) are also consistent with the predicted effect of longer methionine depletion to activate GCN2, attenuate protein synthesis and enhance BiP modification in the longer labeling periods used to generate the older, SubA-resistant BiP.

We applaud the reviewer’s erudition in flagging these issues and much as we are tempted to incorporate these feats of mental gymnastics into the burgeoning revised manuscript, we think it best to resist and thereby refrain from over- fitting our data to prevailing concepts of BiP modification. The manuscript before you is the product of a struggle to strike an intelligent balance in this regard.

3) The authors indicate that Ca++ depletion was shown earlier to interfere with BiP's ability to engage in substrate binding. If the basis of BiP's oligomerization is through its binding to its linker as a substrate, how can the oligomeric state persist upon Ca++ depletion? The attempt to explain these findings in the manuscript is not clear.

This is a very important point (see response to point 4 of reviewer 1). We have incorporated an explanatory paragraph into the Discussion reproduced below:

“We do not understand how calcium depletion shifts the equilibrium in favor of the oligomer. In vitro, there is no measureable effect of calcium on BiP oligomerization, in either the ATP state or ADP state or in the presence of ERdJ3 or the exchange factor GRP170 (not shown), yet in vivo, ER calcium depletion promotes such oligomerization. This discrepancy could be resolved by an ER-localized J-factor whose affinity for BiP as a substrate is increased selectively as [Ca^+2^] _ER_ declines. Given that the affinity of Hsp70 proteins for their substrates is governed by J-protein driven non-equilibrium mechanisms (15), such a scenario could account for BiP favoring itself as a client (oligomerizing) over other clients, as [Ca^+2^]_ER_ declines.”

[Editors’ note: further revisions were requested before acceptance, as outlined below.]

3 September 2015: What follows is a detailed point-by-point discussion of all the reviewers’ comments and revisions they have engendered. Here we should like to list the new experiments:

1) To address the relative roles of ATP binding versus hydrolysis we have examined the effect of ATP on the oligomeric state of a mutant BiP^T229A^ that is defective in hydrolysis (PMID: 1835085, 7592894). Our findings (presented in revised Figure 2—figure supplement 2) confirm that BiP^T229A^ is indeed defective in hydrolysis (panel A) but go on to show that it retains the ability to respond to ATP by de-oligomerising (Panels B & C).

In this experiment we have also measured the effect of ADP and of a non-hydrolysable ATP analog, ATP-gamma S on the oligomeric state of BiP (Figure 2—figure supplement 2). As expected, ADP does not dissociate the oligomers. Consistent with previous experiments in which ATP gamma S has proven to be a weak regulator of allosteric transitions in DnaK (PMID: 8413631), we find that ATP gamma S is also relatively inert.

Together this line of experimentation points to a crucial role for ATP binding (not hydrolysis) in the disassembly of oligomers and showcases the sensitivity of BiP to details of the nucleotide.

2) To further address the impact of the ADDA mutation in the interdomain linker on the affinity of the linker for BiP’s binding site we compare the ability of an established BiP binding peptide [HTFPAVL], the wildtype interdomain linker [DTGDLVLLD] and the ADDA mutant derivative [DTGDADDAD] to compete for BiP oligomerization in vitro. Both the established BiP binding peptide and the wildtype interdomain linker peptide competed effectively, whilst the ADDA mutant linker peptide was without effect (revised Figure 2).

These observations directly support the conclusion (inferred from the effects of the ADDA mutation on oligomerisation in vivo and in vitro and from the cross-linking studies) that the wildtype interdomain linker is a substrate for BiP and that the ADDA mutation, which abolishes oligomerization in vitro and in vivo achieves this by impaired binding to BiP’s substrate binding site.

3) The mass of the complexes resolved by size exclusion chromatography has been assessed by multi-angle lights scattering (MALS, a method orthogonal to migration through a gel filtration matrix) (revised Figure 2—figure supplement 1). MALS validates the mass of complex I as that of a BiP monomer, complex II a dimer and complex III a trimer and confirm that these are distinct oligomeric species, as opposed to alternative conformations of a distinct species. As the quantization of BiP into discrete species follows a similar hierarchy on size exclusion chromatography and on native gels, we believe the insight afforded by MALS on the identity of the former is also informative in regards to the latter.

Essential revisions:

The following points need to be considered.

*1) The authors include new data with mutants T37G and G226D and photocrosslinking experiments to prove the role of the interdomain in oligomerization. This still does not help to explain any possible allosteric effects of ADDA mutant itself. Is there a reason why ADDA was not included in*
Figure 5
*to be compared with WT and other mutants side by side?*

Figure 5 reports on persistent protection of the interdomain linker from SubA digestion in the presence of ATP of the BiP^T37G^ and BiP^G226D^ mutants, compared with BiP^WT^, thereby revealing a correlation between their persistent oligomerization in the presence of ATP (lanes 6 and 8 of Figure 5) and the persistent protection of the linker. Subjecting the ADDA mutant BiP to SubA digestion in the same experiment would be uninformative as a test of its conformational transitions, as the ADDA mutation abolishes (*in cis*) the ability of BiP to serve as a substrate for SubA (see Figure 3 and PMID 17024087; Figure 5 therein).

To cope with this limitation we have employed other proteases to probe allosteric transitions in BiP^ADDA^ (Figure 5—figure supplement 1 and Figure 5—figure supplement 2) and these dissociate the allosteric effects of the ADDA mutation from its effects on oligomerization. The conclusion that the ADDA mutation exerts its effects on oligomerization because it has low affinity for BiP’s peptide binding site is further supported by a new experiment (suggested by a reviewer) comparing the ability of an established BiP binding peptide [HTFPAVL], the wildtype interdomain linker [DTGDLVLLD] and the ADDA mutant derivative [DTGDADDAD] to compete for BiP oligomerization in vitro. Both the established BiP binding peptide and the wildtype interdomain linker peptide competed effectively, whilst the ADDA mutant linker peptide was without effect (revised Figure 2).

*2) Since calcium has no effect* in vitro*, is it possible that what the authors are considering Bip oligomers* in vivo *are in fact Bip in complex with other endogenous substrates? This possibility needs to be considered.*

We believe that the experiments shown in Figure 7 and its supplements, demonstrates the dominant contribution of BiP oligomers to the discrete species that stand out above the heterogeneous signal of BiP immunoreactivity in native gels of cell lysates. This conclusion in no way diminishes the fact that BiP also enters into heterogeneous complexes with many other clients, which likely account for the background signal. This is now explicitly stated in the preamble to Figure 7 (subsection “Linker engagement at the SBD as the basis for BiP oligomerization in vivo 3” of the revised manuscript).

3) The authors failed to examine effect of other nucleotides (AMP/ADP/ATPγS/GTP/GDP, etc), invalidating the claim that Bip aggregate formation is dependent on ATP concentration. Quote here, “As the reviewer well knows, ATP and ADP are expedients to shift the monomer oligomer equilibrium.” It is unclear what this statement means regarding the requirement for nucleotide and ATP hydrolysis in the oligomerization. Is the ATPase activity important for oligomer-monomer conversion?

To address this valid concern, the relative roles of ATP binding versus hydrolysis have been examined by comparing the effect of ATP on the oligomeric state of wildtype with that of a mutant BiP^T229A^ that is defective in ATP hydrolysis (PMID: 1835085, 7592894). Our findings (presented in revised Figure 2—figure supplement 2) confirm that BiP^T229A^ is indeed defective in hydrolysis (panel A) but go on to show that it retains the ability to respond to ATP by de-oligomerising (panels B & C).

We have also measured the effect of ADP and of a non-hydrolysable ATP analog, ATP-gamma S, on the oligomeric state of BiP (Figure 2—figure supplement 2). As expected, ADP does not dissociate the oligomers. Consistent with previous experiments in which ATP gamma S has proven to be a weak regulator of allosteric transitions in DnaK (PMID: 8413631), we find that ATP gamma S is also relatively inert.

Together this line of experimentation points to a crucial role for ATP binding (not hydrolysis) in the disassembly of oligomers and showcases the sensitivity of BiP to details of the nucleotide. We thank the reviewer(s) for this insightful critique.